

# Fast time response measurements of particle size distributions in the 3-60nm size range with the Nucleation Mode Aerosol Size Spectrometer

Christina Williamson[1,2], Agnieszka Kupc[1,2], James Wilson[3], David W. Gesler[4], J. Michael Reeves[5], Frank Erdesz[1,2], Richard McLaughlin[1], Charles A. Brock[2]

[1]Cooperative Institute for Research in Environmental Sciences, University of Colorado, Boulder, CO

[2]Chemical Sciences Division, National Oceanic and Atmospheric Administration Earth System Research Laboratory, Boulder, CO, 80305-3337, USA

[3]Department of Mechanical and Materials Engineering, University of Denver, Denver, CO, 80208-177, USA

[4]St. Mary's Academy, Englewood, CO 80113, USA

[5]Earth Observing Laboratory, National Center for Atmospheric Research, Boulder, CO, 80301

*Correspondence to*: Christina Williamson (christina.williamson@noaa.gov)

**Abstract.** Earth's radiation budget is affected by new particle formation (NPF) and the growth of these nanometer-scale particles to larger sizes where they can directly scatter light or act as cloud condensation nuclei (CCN). Large uncertainties remain in the magnitude and spatiotemporal distribution of nucleation (less than 10 nm diameter) and Aitken (10-60 nm diameter) mode particles. Acquiring size-distribution measurements of these particles over large regions of the free

troposphere is most easily accomplished with research aircraft.

We report on the design and performance of an airborne instrument, the nucleation mode aerosol size spectrometer (NMASS), which provides size-selected aerosol concentration measurements that can be differenced to identify aerosol properties and processes or inverted to obtain a full size-distribution between 3 and 60 nm. By maintaining constant downstream pressure the instrument operates reliably over a large range of ambient pressures and during rapid changes in

altitude, making it ideal for aircraft measurements from the boundary layer to the stratosphere.

We describe the modifications, operating principles, extensive calibrations, and laboratory and in-flight performance of two NMASS instruments operated in parallel as a 10-channel battery of condensation particle counters (CPCs) in the NASA Atmospheric Tomography Mission (ATom) to investigate NPF and growth to cloud-active sizes in the remote free troposphere. An inversion technique to obtain a size distributions from the discrete channels of the NMASS is described and

evaluated.



Concentrations measured by the two NMASS instruments flying in parallel are self-consistent and also consistent with measurements made with an Optical Particle Counter. Extensive laboratory calibrations with a range of particle sizes and compositions show repeatability of the response function of the instrument to within 5-8% and no sensitivity in sizing performance to particle composition. Particle number, surface area, and volume concentrations from the data inversion are

determined to better than 20% for typical particle size distributions. The excellent performance of the NMASS systems provides a strong analytical foundation to explore NPF around the globe in the ATom dataset.

# 1 Background

Particles play important roles in chemical and physical processes in the atmosphere: they provide sites for heterogeneous

reactions (Ravishankara, 1997); they serve as nuclei for the formation of clouds; and they directly and indirectly affect the Earth's radiation budget (Solomon and IPCC Working Group Science, 2007). Many primary particles, those directly emitted into the atmosphere in the solid or liquid phase, affect the radiation budget by acting as cloud condensation nuclei (CCN) or directly scattering or absorbing sunlight. However, secondary particles, those formed by nucleation from the gas phase in the atmosphere, often dominate both aerosol-cloud and aerosol-radiation interactions (Kulmala et al., 2004). By number, the

majority of the particles present in the troposphere in most environments have diameters <100 nm, and substantial fractions of the total particle surface area and volume sometimes lie within this size range (Clarke and Kapustin, 2002).

Once secondary particles have grown to diameters greater than about 50 nm, they often serve as CCN under conditions of water supersaturation common in the lower troposphere (Seinfeld and Pandis, 2006). However, the uncertainty in the contribution of secondary particles to global CCN abundance is very high, with recent estimates ranging from 5% (Wang

and Penner, 2009) to 60% (Yu and Luo, 2009). This uncertainty stems, at least in part, from poorly constrained new particle formation (NPF) mechanisms in the free troposphere. These mechanisms determine not only the nucleation rate (which may only be of minor importance (Westervelt et al., 2014)), but more importantly the spatiotemporal distribution of freshly nucleated particles, which directly affects the number and distribution of secondary CCN (e.g., Merikanto et al., 2009). Measurements of the spatio-temporal distribution of nucleation mode aerosol in the atmosphere can be used to infer the

contribution of different NPF mechanisms and condensable vapors to formation and growth of these particles (Yu et al., 2010;Kazil et al., 2010) . Understanding how much gas-phase species from anthropogenic origins contribute to these processes in comparison to species that have natural origins and may have been present in the pre-industrial era, will enable us to better constrain aerosol-cloud interactions in the pre-industrial atmosphere (Carslaw et al., 2017). Measuring newly formed particles and their growth in pristine areas of today's atmosphere can help us understand the contribution of these

processes to the Earth's pre-industrial radiation budget, and therefore improve our estimates of aerosol radiative forcing. Since new particles form at initial diameters around 1 nm (Kulmala et al., 2000) they must undergo significant growth to become CCN, increasing their diameter by ~50 times and their mass by 5 orders of magnitude. The likelihood that a particle



will be lost by coagulation with larger particles on this journey is high, especially given the high diffusivity of such small particles. Pre-existing larger particles both compete with NPF for condensable vapors, and remove newly formed particles via coagulation. Therefore, the location where nucleation takes place, relative to other sources of particulate matter, and the removal and growth processes they are subjected to, play a large role in the overall importance of secondary aerosols to CCN

abundance and the Earth's radiation budget. It is therefore important to measure the distributions of nucleation (<10 nm diameter) and Aitken (10-60 nm diameter) mode particles to identify regions of NPF and track the process of growth to CCN sizes that they must undergo in order to affect the Earth's radiation budget. A primary measurement is the distribution of particle number concentration as a function of diameter - the particle size distribution. From this measurement, the aerosol surface area and volume concentrations can be calculated, and cloud-nucleating activity can be estimated.

We are measuring the global distributions of aerosols on the Atmospheric Tomography (ATom) mission (https://espo.nasa.gov/missions/atom/). ATom is designed to survey the composition of the troposphere over the remote Pacific and Atlantic oceans from the Arctic to the Antarctic. This is accomplished by flying the NASA DC-8 research aircraft equipped with a comprehensive suite of gas-phase and aerosol instruments while making nearly continuous profiles of the troposphere from ~ 0.15 to >12 km in altitude.  ATom is composed of four sets of flights flown in July-August 2016,

January-February 2017, September-October 2017, and April-May 2018. Each set of flights ranges from California to Alaska to New Zealand, across the Southern Ocean to southern Chile, northward through the central Atlantic to Greenland, and across the Arctic to Alaska and then California.

One of the goals of ATom is to map the spatial distribution of newly formed particles, as well as those large enough to act as CCN. These measurements are being used to constrain NPF mechanisms used in global chemistry-climate models, and to

evaluate loss and growth mechanisms that influence the abundance and spatial distribution of cloud-active particles. These tasks require accurate and precise measurements of the aerosol size distribution spanning 3-1000 nm in diameter, which in turn require a coordinated and inter-calibrated set of in-situ instruments onboard the aircraft.  In this paper, we describe the operating principles, calibration, and laboratory and in-flight performance of the NMASS instruments used to measure the size distributions of the nucleation and Aitken modes during ATom. The optical particle counters used to measure the

accumulation-mode aerosol size distribution are described in detail by Kupc et al. (2017). The inlet, sampling system, altitude-dependent corrections for diffusional losses, and methodology to combine the different instruments are described by Brock et al. (manuscript in preparation, 2018), along with comparisons between different instruments for measuring aerosol size distribution and abundance during ATom.

Plumes and layers of nucleation and Aitken mode aerosol of vertical thickness around 100 m have frequently been observed

in the free troposphere (Kupiszewski et al., 2013;Schroder et al., 2000;Petzold et al., 1999), and yet questions remain about the ultimate fate of the associated particles in the atmosphere, especially whether they grow and are transported to sizes and locations, respectively, to have significant effects on the radiation budget. Since modern, large research aircraft generally operate at airspeeds above 100 m s$^{-1}$ and descent at rates of >10 m s$^{-1}$, fast-response particle sizing and concentration measurements are needed to optimize the study of these phenomena and variability in the background atmosphere. Our



instrument for measuring the nucleation and Aitken modes, the nucleation mode aerosol size spectrometer, or NMASS, uses 5 condensation particle counters (CPCs) operating at a fixed, reduced internal pressure to provide fast response (1 Hz) size distribution measurements between 3 and 60 nm over a range of ambient conditions. As far as we are aware, no other extant instruments can continuously measure the size-distribution of particles over this size range at 1 Hz in a configuration suitable

for airborne use in highly inhomogeneous regions of the atmosphere. The fast integrating mobility spectrometer (Wang et al., 2017) measures from 8 to 600 nm with 1s resolution, but detects only the fraction of the aerosol that is charged in an ionizer, limiting its sensitivity. Multistage electrical mobility instruments using electrometers as detectors typically require very high concentrations (on the order of a few thousand particles per $cm^3$) to achieve fast-response detectability, limiting their application in atmospheric measurements. Nano-scanning mobility particle spectrometers (SMPS) typically take at least 30 s

to measure a size distribution over this size range, over which time an aircraft such as the NASA DC-8 will have travelled 3 km laterally and possibly ~0.25 km vertically. The NMASS instrument provides fast time response with the tradeoff of lower size resolution than an SMPS.

While the NMASS instrument has been used on research aircraft since 1999, in the stratosphere (Borrmann et al., 2010;Lee et al., 2003), and troposphere (Brock et al., 2000;Schroder et al., 2000;Petzold et al., 1999), a comprehensive description of

the instrument and its uncertainties has not been published. In the following sections, we describe the principle of operation of the NMASS, laboratory studies describing its sensitivity to particle number concentration, size, and composition, and the numerical inversion to produce a size distribution from the discrete CPC measurements. Finally, we describe the operation of two NMASS instruments sampling in parallel during ATom mission, which together provide ten channels of 1-Hz size discrimination between 3 and 60nm.

**2 Instrument Description**

**2.1 General Concept**

The NMASS is comprised of 5 parallel CPCs operating at an internal pressure of 120 hPa (Fig. 1). Each CPC detects particles above a different minimum size, determined by the maximum vapour supersaturation encountered by the particles. Operated in parallel, the CPCs provide continuous concentrations of particles in 5 different cumulative size classes between

3 and 60 nm. Knowing the response function of each CPC, numerical inversion techniques can then be applied to recover a size distribution from the continuous concentrations while taking into account the non-ideal response function of each channel.





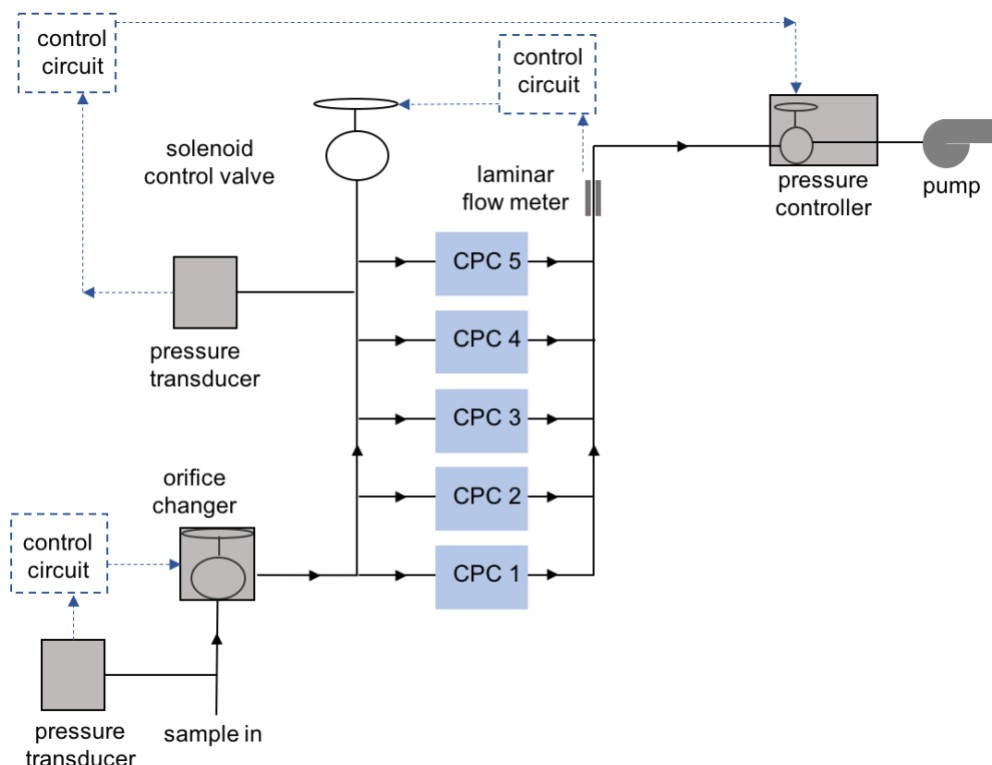

**Fig. 1 Schematic of the NMASS layout and flow system.**

With pressure and flow kept constant, the supersaturation in each CPC is determined by the absolute temperature of the saturator and by the difference in temperature between the saturator and the condenser.  The 5 CPCs are set to different

minimum detection sizes by varying this temperature difference while the saturator temperature of each unit is held constant. The sizing limits are constrained by diffusion losses within the instrument and practical limits to the degree of thermal control required for nucleating large particles at relatively low supersaturations.

The design of this instrument owes much to previous efforts to study and improve the performance of CPCs. In particular Saros et al. (1996), Wiedensohler et al. (1994), and Mcdermott et al. (1991) have demonstrated that the supersaturation

within a CPC can be effectively manipulated to control its detection efficiency as a function of particle diameter (see McMurry (2000) for a history of CPC development). The differences in detection efficiencies among different, individual CPCs have been used to determine the concentrations of particles over one or two size ranges, particularly to identify the presence of an ultrafine (<10 nm) mode (e.g., Clarke and Kapustin (2002), Schroder et al. (2002), Heintzeberg et al. (2001). These studies have not attempted to recover continuous size distributions from the discrete CPC measurements. Gallar et al.

(2006) demonstrated that a size distribution could be recovered by inverting CPC concentrations measured by stepping the supersaturation of a single CPC over a large number of settings. CPC batteries have been examined by Kulmala et al. (2007) from the standpoint of using the composition-depending sizing of different working fluids to understand the chemical composition of atmospheric aerosols, but not as a tool for measuring aerosol size distributions.



The NMASS instrument operates the 5 embedded CPCs using a single integrated data acquisition and control system, flow and temperature regulation systems and power supplies, and an external pump and pressure controller. The physical layout of the instrument is constrained by space and weight limitations for the operation on stratospheric aircraft (it was designed for use on the NASA ER-2 high altitude research aircraft), the dissipation of heat, the need to limit particle losses due to

diffusion, minimization of electronic noise, and accessibility to components for maintenance and repair. Because the instrument was originally designed for autonomous operation in wing-mounted aircraft pods, there is no integral display or user interface. The instrument has a mass of 35 kg in flight-ready configuration and requires an external 8 kg pump. Dimensions are approximately 720 mm long by 360 mm wide by 390 mm high. The instrument consumes <400 W of power, including pumps, and operates from 28 VDC power (18-36 VDC range), which is usually abundant on research aircraft. The

two NMASS units used on ATom and described in this paper show no substantive differences, but will be referred to throughout the paper as NMASS 1 and NMASS 2 (NM1 and NM2 in figures), to differentiate between them.

**2.2 Design of CPC modules**

The Wilson, Hyun, and Blackshear (1983) CPC designed for the NASA ER-2 and WB-57 high-altitude research aircraft provided the concept for confining the aerosol to the center streamline in the condenser and established a condenser

geometry that functioned at pressures from 400 hPa to 40 hPa.  These features were incorporated in the NMASS CPCs which have been operated at pressures from 60-120 hPa depending on the altitude range of the particular aircraft.
In the NMASS, air enters a single inlet through a pressure-reducing orifice and is then carried to each of the CPCs (Fig. 1). The flow entering each CPC is split into two branches (Fig. 2). The first branch passes through a filter (Model DIF-BK40, Headline Filters Ltd., Aylesford, UK) to remove particles, and then through a saturator controlled at 39 °C where vapour

from a perfluorinated organic compound (perfluorotributylamine, Fluorinert FC-43, 3M Specialty Chemicals, St. Paul, MN, USA) diffuses into the airstream to reach the saturation vapour pressure. The vapour-laden flow is carried to a vertical, cylindrical extension of the saturator, where the aerosol flow is introduced from a capillary coaxially into the center streamline. After a short vertical section in which the Fluorinert vapour diffuses radially into the central aerosol flow, the combined flows pass into a cylindrical condenser maintained at a temperature below the saturator temperature. The rapid

temperature drop causes the vapour to become supersaturated and nucleate into droplets onto the particles in the aerosol flow. The droplets grow by condensation and are sensed by an optical detector. The count rate of droplets can be converted into a particle concentration by dividing by the volumetric flow rate of the aerosol, which is determined from the measured pressure drop in the capillary. Key dimensions of the NMASS are given in the Supplementary Material Section A.



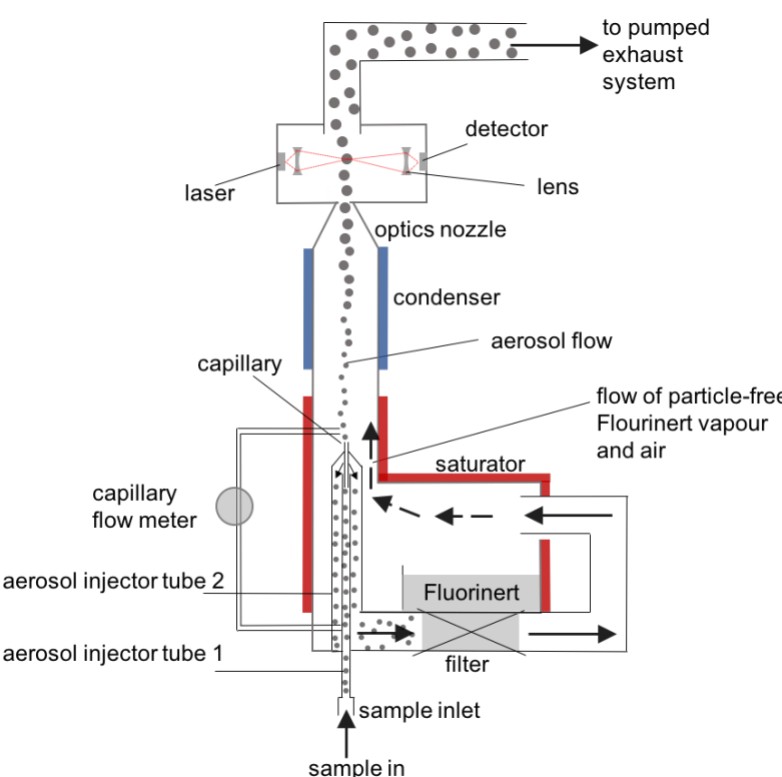

**Fig.2 Schematic of a CPC unit of the NMASS. The sample flow is split between aerosol flow and sheath flow. Aerosol flow is selected from the centre of the sample flow stream and passed through a capillary to reduce loses. Sheath flow is passed through a filter to remove particles, over a warm Fluorinert bath to pick up the vapour and then into the condenser. In the condenser, Fluorinert vapour in the sheath flow diffuses into the aerosol flow and condenses onto particles, growing them to optically detectable sizes. The grown particles then pass through the optics block, consisting of a laser aligned with a lens, beam-block and detector. The measured flow across the capillary and the counts in the optical detector are used to calculate the concentration of the particles. Dimensions are given in the Supplementary Material table S1.**

**2.3 Working fluid choice**

10   The most commonly used working fluid for CPCs is n-butanol. In the NMASS, Fluorinert FC-43 is used instead of n-butanol

for three primary reasons: 1) it offers good thermodynamic characteristics, including a vapour pressure of 4.5 hPa at 39°C

and low mass diffusivity; 2) it is not flammable, toxic or odorous (although it has a high global warming potential, GWP

>5000 (Electronics, 2009)); and 3) it is an extremely inert compound that is not likely to interact chemically with

atmospheric particles, minimizing sizing biases that can occur with n-butanol (e.g. (Weber et al., 1993;Hanson et al., 2002)

15   One positive aspect of Fluorinert's thermodynamic performance is that it requires less precise thermal control than do other

commonly used CPC working fluids. Since the NMASS relies on differencing concentrations from the CPCs to get a size

distribution, a stable diameter response of each CPC is important and requires precise thermal control. In Fig.3, we compare

the temperature dependence of the Kelvin diameter $D^*$, or critical diameter, that refers to the smallest particle size onto





which a given working fluid will condense at a given supersaturation (determined, for steady flow and pressure, by the temperature difference between the saturator and condenser), for n-butanol, diethylene glycol and Fluorinert. For a given $D^*$ and temperature T, the slope $dD^*/dT$ is smaller for Fluorinert than for the common CPC working fluids n-butanol or diethylene glycol. The choice of Fluorinert thus reduces sizing variations caused by uncontrolled fluctuations in temperature.

Details of these calculations are given in the Supplementary Material, section B.

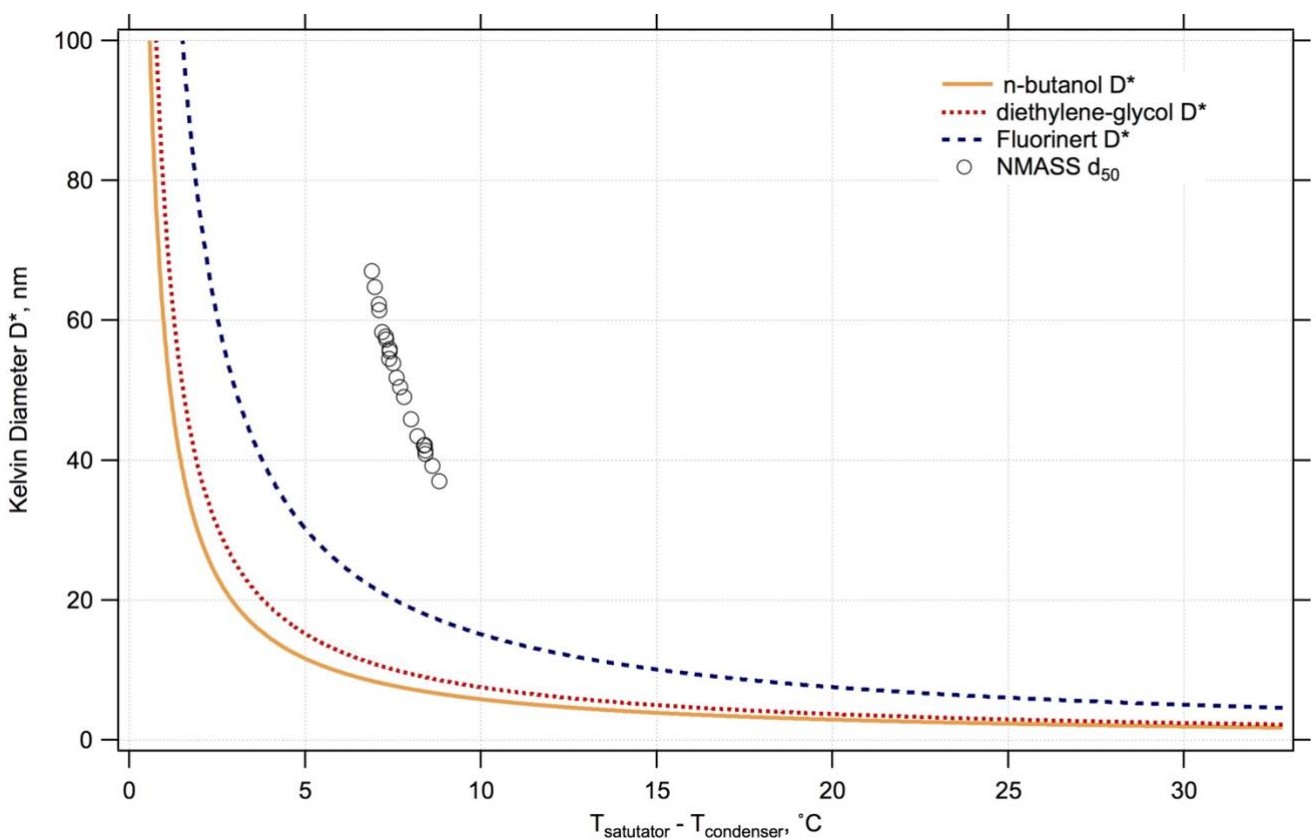

**Fig.3 Kelvin diameter, or critical diameter, $D^*$ as a function of difference in temperature between saturator and condenser, for n-butanol (used in many commercial CPCs such as the TSI-3776 (Hermann et al., 2007)), diethylene glycol (used in commercial and custom-built two-stage CPCs such as the Airmodus Particle Size Magnifier (Vanhanen et al., 2011)) and Fluorinert FC-43 (used in**
**the NMASS CPCs). The saturator temperature is 34.8°C. For a given $D^*$ the slope of the curve for FC-43 is less than for n-butanol or diethylene glycol. The measured diameter of 50% detection efficiency, $d_{50}$, for an NMASS CPC is also shown as a function of temperature, as discussed in section 3.2. The NMASS CPC $d_{50}$s are larger than the theoretical Kelvin diameter for Fluorinert because of particle losses in the instrument and inhomogeneity in the supersaturation achieved within the condenser. The range of $d_{50}$s shown here, around 40-60nm, are on the steep part of the diameter curve. This limits the largest $d_{50}$ that can be achieved with**
**the NMASS because, in this region, a small variation in temperature difference causes a large variation in $d_{50}$, making the detection efficiency unstable.**



## 2.4 Thermodynamic control

The supersaturation reached in the condenser of each CPC of the NMASS is a function of the heat and mass transfer within the condenser, which depends on pressure and flow rate as well as temperature. The dependence of the maximum supersaturation on pressure is complex and difficult to model (Stolzenburg and Mcmurry, 1991), especially since not all

needed thermodynamic properties of Fluorinert such as mass diffusivity in air are known. Since the response of each CPC varies with ambient pressure, allowing the instrument pressure to vary with altitude on aircraft campaigns would require a large number of calibrations for the different pressure conditions, and parameterizations to characterize how the instrument response varies with pressure. To simplify calibration and avoid the related uncertainties with this parameterization, the CPCs within the NMASS are maintained at a constant internal pressure.

Because the internal pressure of the NMASS must be below ambient pressure at all times, the choice of this pressure depends on the minimum anticipated ambient pressure. Diffusional and inertial particle losses are enhanced at low pressure, so it is desirable to use the highest practical instrument pressure. The NMASS was originally designed for operation on stratospheric research aircraft such as NASA's ER-2 and WB-57F; when operated in the stratosphere an internal pressure of 60 hPa is used. For lower altitude measurements, aboard tropospheric aircraft such as the NASA-DC8 and NOAA's WP-3D,

an internal pressure of 100 or 120 hPa has been used. Internal pressure is maintained by sampling through a thin-plate orifice at the inlet to the instrument and controlling the downstream pressure using a pressure controller and external pump. The response of each CPC unit depends on the flow rate as well as the pressure, so the volumetric flow rate through the CPCs must be kept constant. Because mass flow through the orifice changes with ambient pressure, this requires an active flow control system. As the aircraft ascends, flow through the orifice becomes insufficient for the CPCs, so a larger orifice must

be switched in place. The NMASS is thus operated with two different sized orifices, 500 and 750 µm, in the troposphere, and with yet a third orifice of 1200 µm in the stratosphere. The desired orifice is selected using a Swagelok SS-44F6 ball valve machined with a second channel perpendicular to the original channel. An orifice is brazed into the exit of each of the channels. This is illustrated in Fig. S1 in the Supplementary Material. Rather than operating as an on/off valve, after this modification when the valve is turned a different orifice is selected. During ATom, the orifice switch occurs at 450 hPa on

ascent and 400 hPa on descent (providing some hysteresis to prevent rapid switching in the event the aircraft is flying level near the switching pressure). A valve actuator (M-series, Hanbay Inc., Pointe-Claire, Quebec, Canada) is used to drive the orifice valve. We minimize particle losses through the pressure reducing inlet with a design based on that of Lee et al. (1993).

Total flow through the NMASS instrument is controlled automatically by adjusting a proportional control valve (Model

248A, MKS Instruments Inc., Andover, Massachusetts, USA) that regulates flow through a bypass line, as shown in Fig. 1. This flow circuit maintains a nearly constant volumetric flow through the CPCs even as changes in upstream pressure alter the volumetric flow downstream of the orifice.

Determining a size distribution by differencing parallel instruments requires precise control of the instrument response. At



relatively low values of supersaturation, small changes in temperature difference between the saturator and condenser can produce large excursions in supersaturation, and thus $d_{50}$ (see Fig.3). The absolute temperature of each saturator is monitored by two high-precision thermistors per channel and maintained by resistive heaters. The power to the heaters is controlled by a custom proportional feedback control circuit. The temperatures of the condensers are maintained by thermoelectric

(Peltier-type) coolers and governed by circuits that control the difference in temperature between the saturator and the condenser.

The counting efficiency of each NMASS channel as a function of diameter can be varied by changing the temperature difference between the saturator and condenser, and varies with internal pressure. We chose to operate the NMASS at ~120 hPa for the ATom mission. Condenser temperature settings were chosen to space the 5 channels of each NMASS

approximately evenly within logarithmic space between 3 and 60 nm. The channels of the two instruments were offset from each other, to provide 10 distinct diameter cut-points, but also allow for nearly complete coverage of the nucleation and Aitken modes should one instrument experience problems in flight.

**2.5 Optical detection and processing**

The Fluorinert droplets nucleated and grown in the condensers are detected with 5 simple optical particle counters which use

near-IR laser diodes as the light source and a forward scattering geometry. The optics blocks and laser diode electronics are modified versions of the Model 3760/3010 detector (TSI, Inc., St. Paul, MN, USA). The scattered light is detected with a photodiode. Custom electronics correct for shifting of the baseline voltage from the photodiode circuit as concentration increases. Analog pulses are converted to TTL-level signals within a shielded enclosure, then routed to 32-bit counter/timer circuits. Total counts in each channel are accumulated over a time period determined by software (typically 1s). The fraction

of time in each second in which particles are occupying the laser beam and the system cannot process another particle (the dead time) is measured using 5 additional 32-bit counter/timers. Corrections are made for dead time by dividing the number of counts by the fraction of measurement time for which the detector electronics are not busy processing. This correction allows ambient concentrations up to $3 \times 10^5$ cm$^{-3}$ of air per channel to be accurately measured. Data acquisition and storage are controlled by a PC/104-bus computer.

**3 Calibration and Laboratory Performance**

**3.1 Calibration methodology**

Laboratory studies were used to determine the counting efficiencies of each NMASS channel as a function of particle diameter. Aerosols were produced with three different methods: 1) limonene ozonolysis, 2) atomization of ammonium sulphate, and 3) atomization of 2-diethylhexyl (dioctyl) sebacate. These methods produce particles of widely differing

composition that can help identify any composition-dependent sizing effects. The dependence of the counting efficiency with size was studied by placing a Boltzmann steady-state charge distribution on the generated particles with a Po-210 neutralizer



and passing them through a nano-DMA to select particles of a single electrical mobility (Fig.4). The fraction of doubly charged particles is small for particles with diameters <100 nm. No neutralizer was included after the nano-DMA, so the particles being detected by the NMASS and CPC were negatively charged. The applicability of this calibration to atmospheric particles, which are mostly neutral, may be affected by the use of charged particles for the calibration.

Stolzenburg and Mcmurry (1991) showed that positively and negatively charged particles have very similar activation diameters in a butanol CPC, and Kuang et al. (2012) and Winkler et al. (2008) found butanol, diethylene-glycol and n-propanol CPCs to have higher sensitivity to charged particles than neutral particles at small sizes (diameters less than 4nm). Therefore, we assume that the effect difference between the performance of the NMASS CPC for charged and neutral particles will be small, and mostly limited the smallest particle diameters. The effect of composition on counting efficiency

is evaluated by comparing response curves from a single channel for the three different particle compositions.

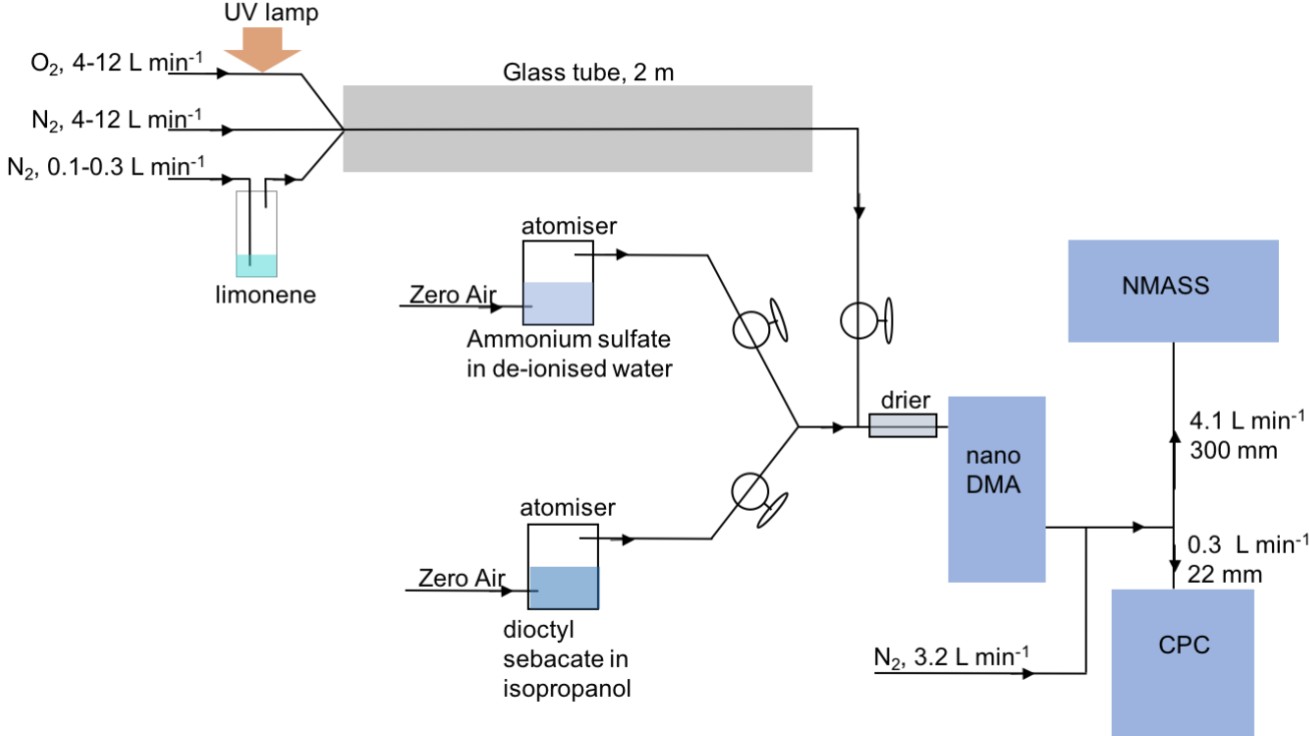

**Fig.4 Diagram of the calibration set up used to characterise the NMASS counting efficiency as a function of particle diameter. The calibration includes different aerosol types such as limonene, ammonium sulphate and dioctyl sebacate particles to test the instrument sensitivity to particle composition.**

We used a nano-DMA column (Model 3085, TSI, St. Paul, MN, USA) in a custom-built DMA system with non-recirculating sheath flow. By varying the sheath flow between 3 and 16 L min$^{-1}$, and adding dilution flow between 0 and 3.2 L min$^{-1}$, we were able to size select particles from 2 to 80 nm whilst keeping the ratio of aerosol flow to sheath flow in the column at



1:10. To select particles with diameters >100 nm a longer custom-built DMA column was used in a DMA system with recirculating sheath flow. Dilution flow was again used to keep the ratio of aerosol to sheath flow at 1:10. We calculated the uncertainty on the diameter of particles selected by the DMA by propagating uncertainties in voltage, pressure, flow and DMA dimensions (see section D of Supplementary Material) using a root-mean-squared method. The overall diameter

uncertainty was dominated by the width of the distribution selected by the DMA and so the full-width-half-maximum of this is reported as the total uncertainty on the particle diameter. We did not account for diffusional broadening of the DMA transfer function.

Two n-butanol-based nano-CPCs (a Model 3776 and Model 3025A, TSI, Inc., St. Paul, MN, USA) were used as reference CPCs. Each of these instruments has a size-dependent response function which was taken into account in the analysis. For

the 3025A CPC, we used the Stolzenburg and Mcmurry (1991) calibration, taking the Kangasluoma et al. (2014) calibration with limonene ozonolysis products into account in the uncertainties. For the 3776 CPC, we used the Hermann et al. (2007) calibration with silver particles, taking into account calibrations with sodium chloride from the same paper and calibrations with sucrose and silver from Mordas et al. (2008) in the uncertainties. These calibrations and uncertainties were applied to all diameters <10nm for 3025A and <7 nm for the 3776. At larger sizes the reference CPCs count particles with ~100%

efficiency.  To avoid the influence of differential diffusion losses between the reference TSI 3025A CPC and the NMASSes, the lengths of the sampling lines going to the CPC and the NMASS were made proportional to the flow rate passing through them.

The efficiency of each NMASS CPC is taken as the ratio of the standard temperature and pressure (STP, taken as 273.16 K and 1013 hPa) concentration measured in the NMASS to that measured in the reference CPC. The concentration of the

NMASS and reference CPCs is calculated as the number of pulses counted by the instrument per unit of time divided by the flow rate, corrected for dead time. This concentration is corrected for pressure and temperature to get the STP concentration. The uncertainty in the NMASS downstream pressure at 120 hPa is ±0.5 hPa, and the CPC internal pressure uncertainty is ±2 hPa. The flow calibrations of the NMASS CPCs have an uncertainty of ±7%, while the reference CPC flow uncertainties are ±10%. The uncertainty on the count rate is $\sqrt{N}$, where N is the number of counts in the averaged interval (1s time intervals

are used for this calibration), for both NMASSes and CPCs. The temperature correction is small enough for uncertainties on it to be neglected. The total uncertainty on the NMASS efficiency is then the sum of the flow, pressure and counting uncertainties in quadrature.

## 3.2 Sensitivity to particle size

The counting efficiency for each channel as a function of diameter for particles produced from limonene ozonolysis for the

settings used during ATom are shown in Fig.5. Fits to the response curves are included to guide they eye. Condenser temperatures and diameters at which each channel detects 10, 50 and 90% of the particles are given in Table 1. For particles >150nm, the counting efficiency begins to drop off with increasing particle size due to impaction losses in the pressure reducer. We characterized these losses using size selected ammonium sulphate particles (Fig.6). Ammonium sulphate was





used for this calibration since it was not possible to generate enough limonene ozonolysis particles of large enough diameter, and, as discussed in section 3.3 the NMASS seems to be insensitive to particle composition. The loss of these particles is likely to be dependent upon pressure upstream of the orifice (Lee et al., 2003). Because the NMASS instrument is always paired with an optical particle counter to explicitly measure the accumulation mode, these impaction losses have no practical

effect on the final size distribution produced by combining the instruments (Brock et al., manuscript in preparation, 2018). The repeatability of each CPC's response function is limited by temperature, pressure and flow control. The variation in 50% detection efficiency diameter with respect to temperature difference between saturator and condenser is shown in Fig.3. The saturator temperature was held constant at 34.8°C while the condenser temperature was varied between 26 .0 and 27.9 °C. The gradient of the Kelvin curve is a determining factor in the NMASS detection efficiency stability. If the curve is too

steep, any small thermal instability will cause a large variation in $d_{50}$. It can be seen in Fig.3. that the theoretical Kelvin curve for Fluorinert FC-43 becomes very steep around 60 nm, therefore we avoid setting any $d_{50}$ above 60 nm in the interest of stability. The flow, pressure and temperature stability achieved in the NMASS are discussed in sections 3.4 and 3.5. Sizing is less accurate and precise the lower the supersaturation achieved in the CPC (corresponding to larger particles), as is seen in Fig. 6.

For a given temperature difference between saturator and condenser, the measured $d_{50}$ in Fig.3 is larger than the theoretical Kelvin diameter. The Kelvin diameter is the minimum diameter at which it is possible for particles to nucleate, while $d_{50}$ is the diameter at which 50% of particles are detected in the instrument; some difference is expected. In particular, it is likely that the NMASS saturator does not fully saturate the sheath flow. As long as the degree of saturation is constant (which it is expected to be since pressure, flow and temperature are constant), the $d_{50}$ of each NMASS channel should also be constant.



**Fig. 5 Counting efficiency of the two NMASSes in the settings used for the ATom mission (downstream pressure at 120 hPa, saturator temperatures of both instruments are set to 39°C, condenser temperature of NMASS 1 are 2.6, 16.2, 21.4, 26.2 and 29.7°C and condenser temperatures of NMASS 2 are 12.2, 13.6, 20.4, 27.2 ad 30.7°C). The calibration (symbols) is done using particles generated by ozonolysis of limonene, and NMASS concentrations are compared to a TSI 3025 or 3776 CPC, with the CPC concentration corrected for its own counting efficiency at each diameter. Lines are fitted logistic functions to guide the eye.**





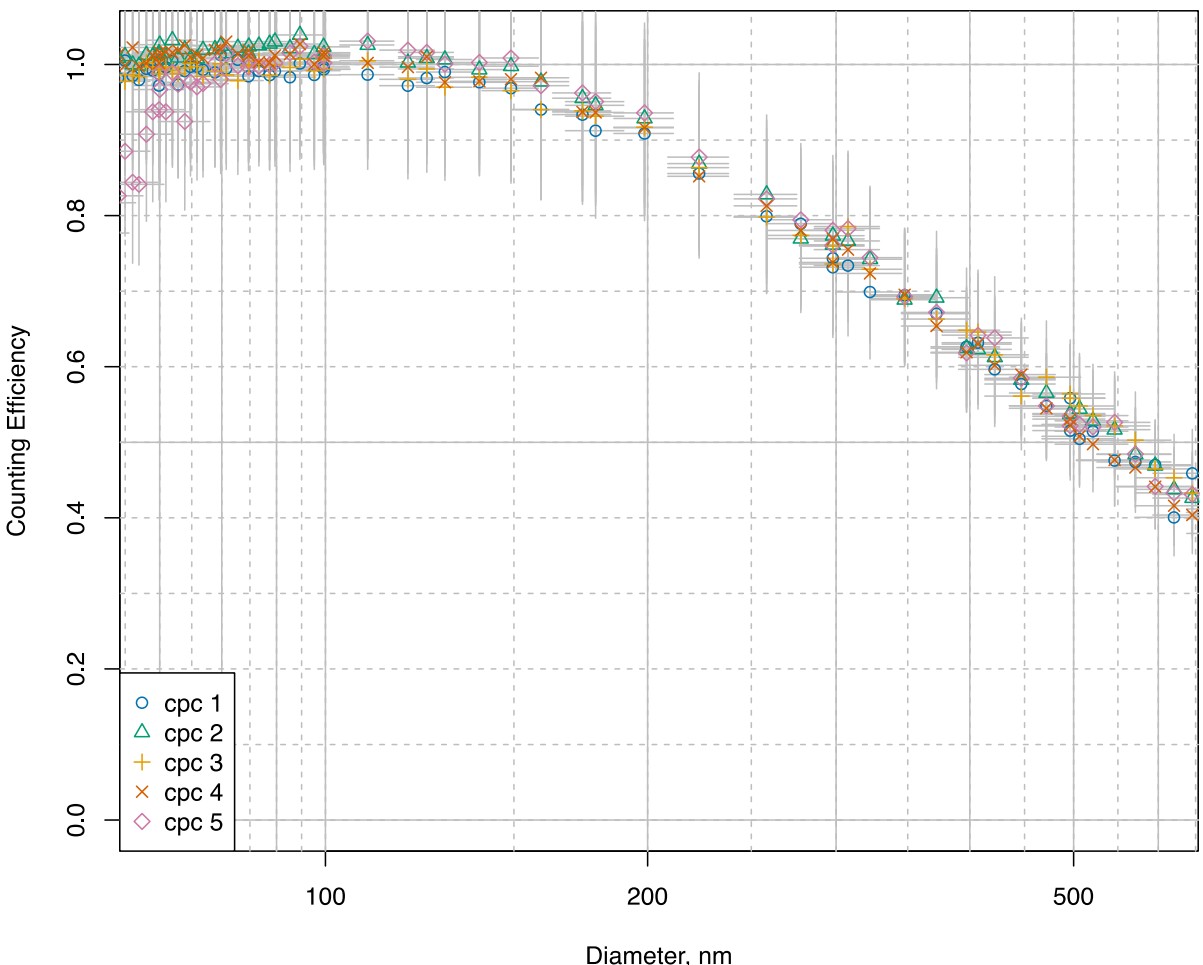

**Fig. 6 Counting efficiency of NMASS 1 against particle diameter for diameters between 65 and 650 nm using atomised ammonium sulphate particles. The decay of counting efficiency with particle diameter at diameters above 150 nm is caused mainly by particle impaction on the orifice at the instrument inlet. At particle diameters above 70 nm, all 5 channels (shown by the different colours and symbols in the legend) have the same counting efficiency. Particle counting efficiency is 100% at 109 nm and drops to 50% at 546 nm.**

### 3.3 Sensitivity to composition

Sensitivity of the NMASS to the composition of the aerosol sample was tested by calibrating with ammonium sulphate and

dioctyl sebacate particles generated using an atomizer, and comparing this to the calibration with limonene ozonolysis

particles. These three compositions were chosen because they can be produced using a flow tube reactor or an atomizer, and

represent a range of different particle compositions similar to those found in the atmosphere. Limonene ozonolysis products





represent particles nucleated and grown with low volatility organic compounds, dioctyl sebacate particles are liquid organic droplets, and ammonium sulphate is representative of aged atmospheric sulphate particles fully neutralized by ammonia. Calibrations with these different compositions show no statistically significant variation between counting efficiency curves (Fig.7). Note that sensitivity to composition was only done for particles >20 nm in diameter since it was not possible to

5   produce smaller atomized particles of ammonium sulphate and dioctyl sebacate.

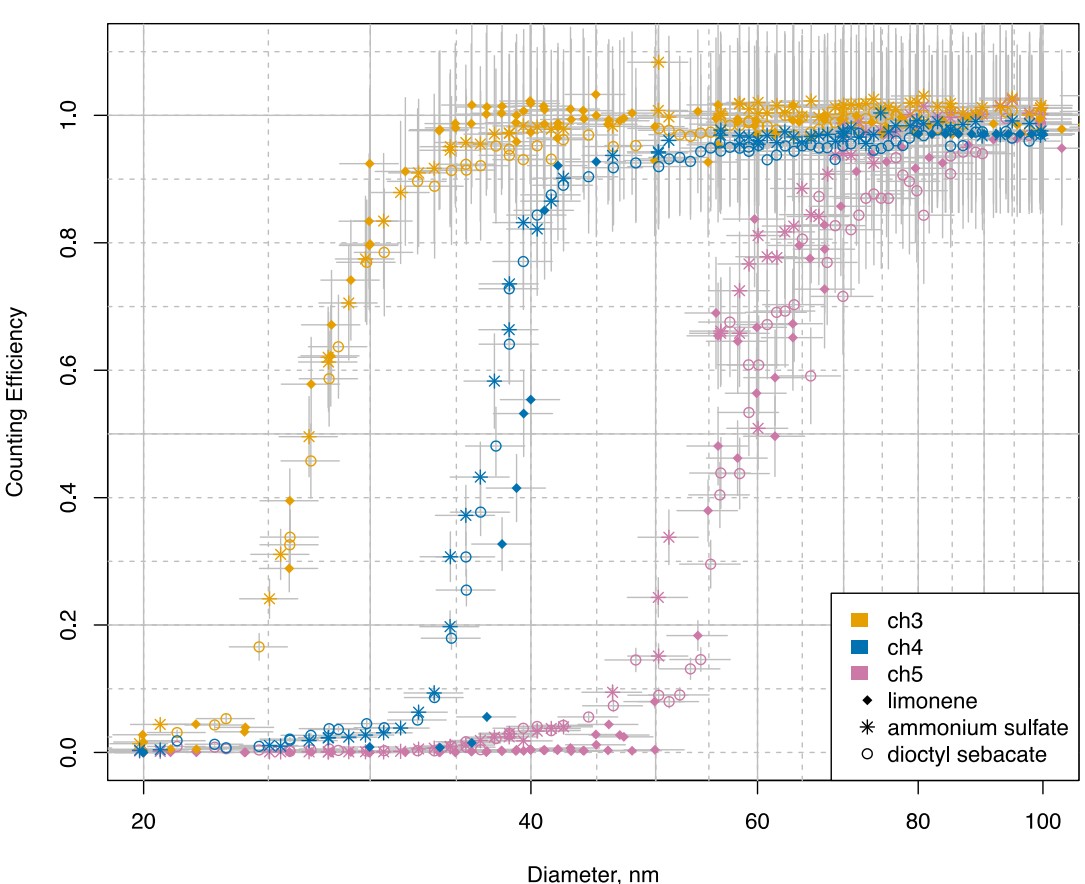

**Fig. 7 Counting efficiency of NMASS 1 as a function of particle diameter for particles of different chemical composition: limonene ozonolysis products (diamonds), atomized ammonium sulphate (stars) and dioctyl sebacate (circles). Only three channels are shown here because it was not possible to produce atomized particles small enough for the two channels with the smallest cut-off**
10  **sizes by.**



### 3.4 Stability with respect to thermal drift

At the highest diameter settings (lowest condenser to saturator temperature difference), the supersaturation is at its lowest and therefore more sensitive to any fluctuations in flow and temperature. In each of the NMASS instruments the highest diameter channel exhibits small fluctuations in cut-off diameter, most likely due to non-uniformities in supersaturation

5   through the condenser block and being on the steepest part of the Kelvin curve here (see Fig. 3). NMASS 2 channel 5 shows the largest drift in $d_{50}$, as illustrated in Fig.8, which shows calibrations from three different days for this channel. The value of $d_{50}$ varies from 36-39 nm, a shift of 8%. Other channels exhibit stability within 5%. The effect of these fluctuations in sizing performance on the accuracy of the size distributions recovered from the NMASS measurements is evaluated using Monte Carlo simulations as discussed in section 4.2.





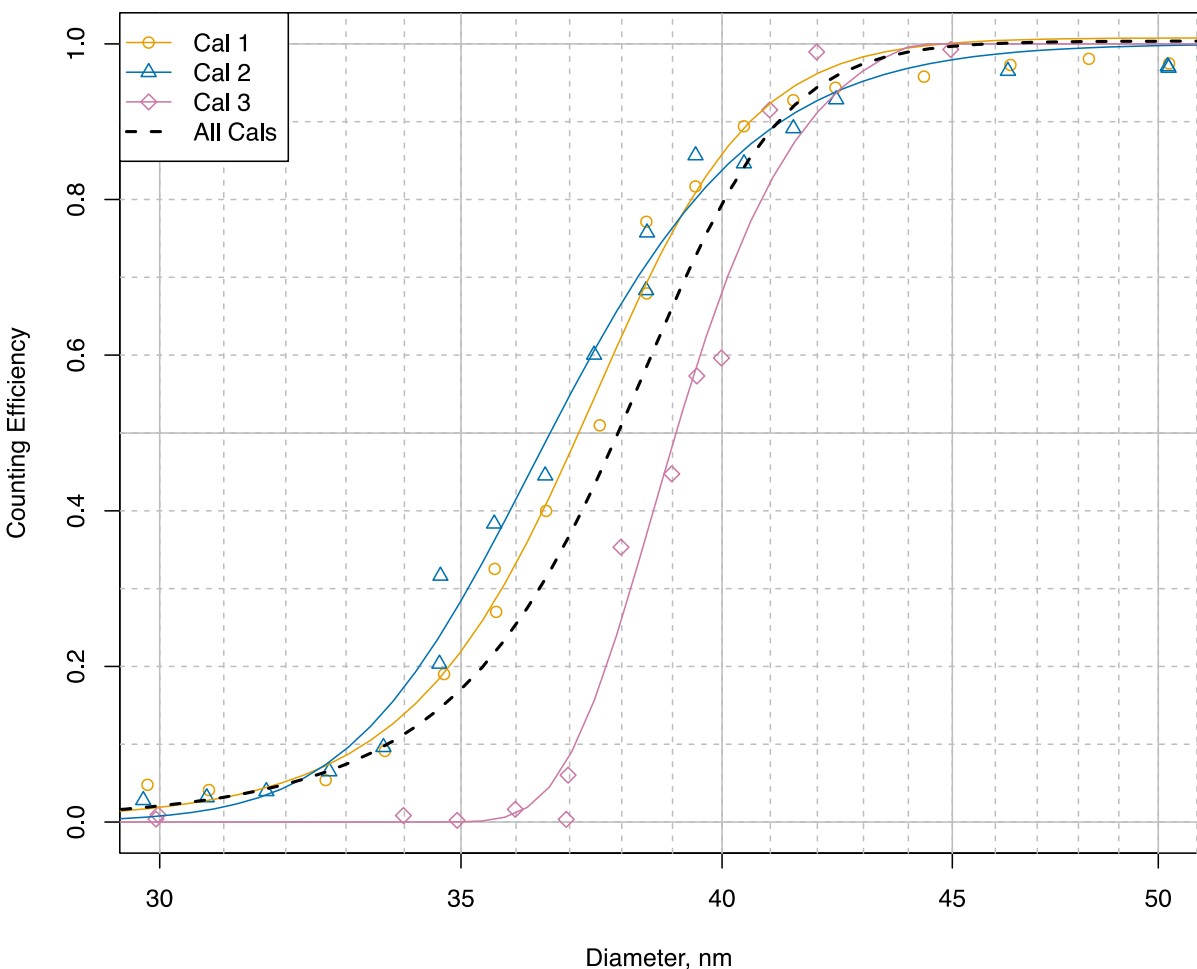

**Fig. 8 Calibrations of NMASS 2 channel 5 in detail. Calibrations on three separate days are shown with fitted logistic functions to guide the eye. This shows instabilities in the supersaturation of the condenser of this channel, with the 50% diameter varying between 36.5 and 39.1 nm. The black dotted line shows the fit to all data, which is used for inverting the NMASS data.**

# 5  4 Numerical Inversion

### 4.1 Methodology

Each of the NMASS channels, $i$, records a single concentration value $X_i$,



$$X_i = \int_0^\infty N(D_p)K_i(D_p)dD_p \tag{1}$$

where $N(D_p)$ is the particle size distribution function, and $K_i(D_p)$ the response function of channel $i$ to a particle of size $D_p$. The inverse problem is to solve for $N(D_p)$ given $K_i(D_p)$ known within experimentally determined uncertainty, and $X_i$

measured with known uncertainty. We use a version of the smoothed Twomey algorithm (Markowski, 1987), a simple inversion technique, to generate a 20-bin-per-decade finite difference representation of $N(D_p)$. Because the equation set is ill-posed, an infinity of possible $N(D_p)$ will produce a given set of $X_i$ (e.g., Wolfenbarger and Seinfeld (1990)). The algorithm uses a nonlinear technique to choose one smooth, non-negative solution that minimizes the discrepancy between the predicted and actual instrument response. Two adjustable parameters, a convergence criterion and a smoothing parameter,

are required for this inversion procedure. The values of these parameters are determined using laboratory experiments with known aerosol size distributions and then fixed for most applications. We use binomial smoothing with two smoothing passes and convergence criterion of 0.1% (Marchand and Marmet, 1983). For the ATom project, the inversion was calculated over a size range from 2.7 nm to 300 nm, constrained by the concentration of particles with $D_p$>63 nm measured by an ultra-high sensitivity aerosol size spectrometer (UHSAS), and combined with the UHSAS and other instruments as

described in Brock et al. (manuscript in preparation, 2018). The pressure variation in the impaction losses in the NMASS (Fig. 9) is not considered in the inversion because the UHSAS data constrain the large end of the inversion, and because the final size distribution is produced by combining the inverted NMASS data with the explicitly measured UHSAS size distribution at $D_p$=67 nm.

### 4.2 Error Propagation

Because of the nonlinear inversion, it is not possible to directly calculate how uncertainties propagate from concentration and sizing errors through to the final size distribution. Instead, we use a Monte Carlo technique to calculate the range in variation of the number, surface area, and volume concentrations for several representative size distributions characteristic of the ATom project. To perform this analysis, eight size distributions that represent a range of those encountered during the ATom flights were simulated using a three-mode lognormal distribution (Table 2). These model distributions were then used to

calculate the expected instrument response; that is, the concentrations that would have been measured by each channel of the NMASS were calculated from the known response functions. These concentrations were then each independently and randomly adjusted by a value falling within the concentration uncertainty as represented by one standard deviation of a Gaussian distribution. Further, the response functions of each channel were similarly independently and randomly adjusted in diameter space using the observed variation in each channel's response during calibration (Table 1). The "perturbed"

concentration measurements and response functions were then inverted to recover $N(D_p)$ using the Twomey-Markowski algorithm. The steps of random perturbation and inversion were repeated 1000 times for each representative size distribution.



The inverted size distribution was compared to the "true" model size distribution for each iteration (Supplemental Material Fig. S2). The relative standard deviation $\sigma_y$ for integrated number, surface, and volume, and for the peak diameter of the size distribution, was calculated as

$$\sigma_Y = \frac{\left(\frac{\sum_{i=1}^{N}(Y_i - Y_{mean})^2}{N-1}\right)^{1/2}}{Y_{true}} \quad , \qquad (2)$$

where $i$ is the Monte Carlo iteration up to a maximum of $N=1000$, $Y$ is the parameter of interest, $Y_{mean}$ the mean $Y$ from all of the inverted size distributions, and $Y_{true}$ the true value of $Y$ from the model input size distribution. Similarly, the mean relative bias $B_y$ was calculated as

$$B_Y = \frac{Y_{mean} - Y_{true}}{Y_{true}} \quad . \qquad (3)$$

Values of these statistics for the eight test size distributions are given in Table 2. The mean magnitudes of $B_Y$ for the integrated number, surface area and volume were <20% for 7 of the 8 size distributions. The seventh size distribution (Case #4), which featured a very low number concentration (23 cm$^{-3}$) with a modal diameter of 4 nm, near the bottom of the NMASS detection range, had a bias of +39% for volume. The $\sigma_Y$ values, representing variability in the number, surface, and volume from the replicates of the inverted size distributions, were <18% except for the same size distribution, which had $\sigma_Y$ values of 24%, 40%, and 91% for number, surface area, and volume, respectively. Note that the biases and variability of the solution for Case #8, which features a very small nucleation number mode (2.5 nm) but a very high number concentration (15000 cm$^{-3}$, representing a more typical case of very recent new particle formation) does not exhibit the variability and biases of Case #4. Thus, for most size distributions encountered in ATom, the uncertainties in integrated number, surface area, and volume generally can be considered to be ±20%. However, uncertainties may be larger in cases where concentrations are low and modal diameters are at the extremes of the NMASS response.

Note that, for particles with diameters lying between the response curves of the first channels of each NMASS instrument (i.e., between ~2.5 and 6 nm; Fig. 5), the inversion is essentially unconstrained. A concentration difference between these two channels could be attributed to large concentrations of ~2.5 nm particles detected with low efficency, or a smaller concentration of ~5 nm particles detected with high efficiency. The number of particles assigned to this size range by the inversion is largely determined by the smoothing parameter, which averages adjacent inverted size bins to produce a smoothly varying size distribution. This ambiguity is evident in the consistently overpredicted concentrations for $D_p < 6$ nm seen in the inverted size distributions (Supplemental Materials Fig. S2). Thus, the NMASS is not well suited for investigating the detailed dynamics of new particle formation and growth. An instrument with much greater size resolution,





such as a nano-scanning mobility particle sizer (nano-SMPS), making repeated measurements of the time-varying evolution of the particle size distribution is needed for such studies. Such measurements of course are not possible on the rapidly moving ATom DC-8 airborne platform, which instead seeks to map the variation of particle characteristics in geophysical and thermodynamic space and use this information to evaluate the plausability of various new particle formation

mechanisms.

## 5 Comparison to SMPS

### 5.1 Method

The system of two NMASSes was compared with an SMPS, the standard technique for ground-based measurements of nucleation-mode particle size distributions. Aerosols were generated by atomizing ammonium sulphate and dried with a

silica gel diffusion drier before entering a custom-built DMA with recirculating sheath flow. The DMA was used to select a narrow size range of particles, and then the sample flow was split between the two NMASSes and an SMPS as shown in Fig.9. The SMPS is made up of a TSI 3050 nano-DMA column and a TSI 3022A CPC. A short section of tube with an inner diameter of 1.59 mm is used after the first DMA to generate turbulence and ensure the sample flow is well mixed before it is split between the instruments. Because the internal pump of the CPC is unable to keep stable flow at 0.3 L min$^{-1}$ given the

pressure drop in this mixing tube, an additional external pump is used, with a valve to set the CPC flow to 0.3 L min$^{-1}$. The first DMA was used to select particles of a given size, which were then sent to the NMASS and SMPS. The SMPS scanned for 10 minutes between 4 and 200nm, and the average NMASS concentrations were calculated over this time. Concentrations from the SMPS and NMASS were inverted using the method described in section 4.1.



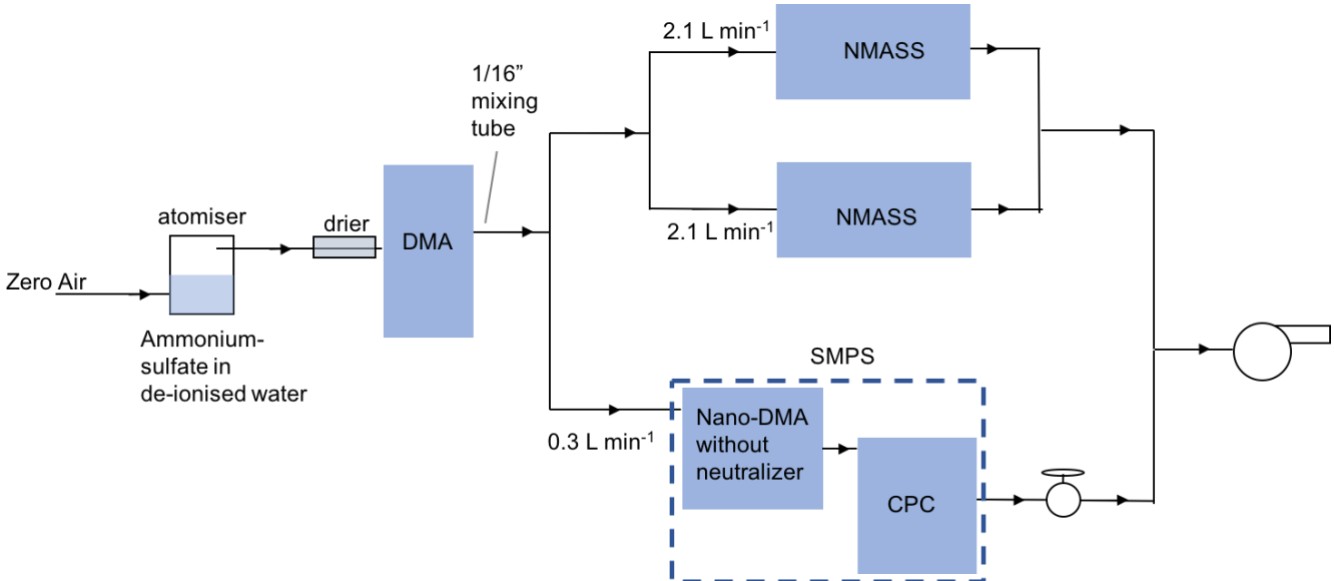

**Fig. 9 Set-up for checking the NMASS inversion against an SMPS. Atomized ammonium sulphate is dried and passed to a DMA, which selects a narrow range of particle sizes. A 1/16" tube is used after the DMA to ensure the sample flow is well mixed (the Reynolds number of the flow, Re > 4000) before splitting it between an SMPS (composed of a nano-DMA coupled with a CPC) and the two NMASSes. The length of tubing to the SMPS and NMASS systems are proportional to the flow through them to ensure the sample experiences the same diffusion losses before being measured. An external pump is used behind the CPC to handle the low upstream pressure caused by 1/16" tubing after the first DMA.**

## 5.2. Results and Discussion

Comparisons of inverted size distributions from the SMPS and NMASS instruments are shown in Fig. 10, with a 20 nm peak selected by the DMA in the upper panel, and 32 nm in the lower panel. For the 20 nm peak, the SMPS peak appears at 19.9nm and the NMASS peak at 21.2 nm, indicating a 6% discrepancy. The largest uncertainties on the SMPS sizing come from the voltage and the time lag correction, which at 20 nm gives a total sizing uncertainty of $\pm$ 24 nm. The peak of the SMPS and NMASS therefore agree within this uncertainty. The NMASS peak is broader then the SMPS because of the intrinsic resolution of the 10 size bins between 3 and 60 nm compared to the high resolution of the SMPS. Smoothing within the NMASS inversion algorithm further broadens the measured size distributions. Such smoothing is appropriate for atmospheric sampling, where geometric standard deviations >1.4 are typical. The integrated total number concentration is 1364 cm$^{-3}$ from the NMASS and 1213 from the SMPS, a discrepancy of 11%. For the 32 nm peak, the larger uncertainties in NMASS 2 channel 5 ($d_{50}$ = 40 nm), become apparent. The standard NMASS calibration places the peak of the distribution at 33.7 nm and the SMPS places it at 32.9 nm. SMPS uncertainties in diameter at 32 nm are $\pm$ 2.4 nm, so this agrees within calculated uncertainty. The integrated total number concentration for the 20 nm peak is 1364 cm$^{-3}$ from the NMASS instruments and 1213 cm$^{-3}$ from the SMPS, a discrepancy of 11%. For the 30 nm peak, the integrated total number concentration is 7604 cm$^{-3}$ from the NMASS instruments and 4529 cm$^{-3}$ from the SMPS, a larger discrepancy of 40%. The



integrated concentration from the inverted NMASS size distribution is consistent with the directly measured NMASS CPC concentrations, while the integrated SMPS size distribution is not.

The effect of variation in sizing of NMASS 2 channel 5 is examined by recalculating the inversion with the range of calibrations from multiple days (Fig. 11). The resulting differences in the inverted size distribution move the peak between

5    33.7 and 37.8 nm. We expect this additional diameter uncertainty of about 12% to show up between 27 and 60 nm, where the measurement from NMASS 2 channel 5 plays a determining role in the particle sizing.

Fig. 10 Comparison of SMPS and NMASS inversions for size distributions produced by atomized ammonium sulphate particles size selected by a DMA. Pane (a) shows a narrow 20 nm particle size distribution. Panel (b) shows a narrow 32 nm particle size

10    distribution produced by atomized ammonium sulphate particles size selected by a DMA. Four different NMASS inversions are shown, using different known calibrations of NMASS 2 ch5 (all other channel calibrations constant), to understand the effect of this instability (discussed in section 3.5) on the 32 nm inversion.



## 6 Performance during flight

Two NMASS instruments have recently been flown on a NASA DC-8 in the boundary layer and free troposphere on the ATom) (Prather et al., 2017), repeatedly profiling between ~200 and >1000 hPa, while travelling between ~80 N and ~60 S in a wide range of conditions. The NMASS instruments were operated at 120 hPa internal pressure, in a flight rack in the pressurized fuselage of the plane. The 1Hz time-resolution, and the use of the orifice and flow control system to maintain constant instrument pressure and flow, made it possible to sample during the constant altitude changes and to measure small-scale features in the aerosol spatial distribution while doing so.

## 6.1 Instrument stability

The stability of the temperatures, flows, and pressures within the NMASSes is critical to maintaining constant instrument response during flight. In Fig.11, we show key parameters for instrument stability: the CPC temperature difference, total CPC flow and instrument pressure. The greatest temperature instability is in NMASS 2 channel 5, where the temperature difference fluctuates by 0.46°C. NMASS 2 channel 2 fluctuates by 0.2°C; all other channels by less than 0.2°C. When the orifices changes at 400 hPa on ascent and 450 hPa on descent, both instrument pressure and CPC flow are perturbed. This perturbation lasts for a maximum of 20 s, and data taken during the perturbation are removed prior to analysis.

The pressure control system with two orifices, described in section 2.4, maintains a total volume flow rate of 122 to 144 cm$^3$ s$^{-1}$ for NMASS 2 and generally between123 and 131 cm$^3$ s$^{-1}$ for NMASS 1 over the ambient pressure range of 225-1100 hPa experienced in ATom flights. On this deployment, the bypass valve in NMASS 1 was too small, so at ambient pressures above 960 hPa, too much flow passed through the CPCs. Data from these times are either discarded, or the uncertainties on the inverted size distributions increased to reflect the additional uncertainty in counting efficiency introduced by excess flow. The downstream pressures occasionally experience fluctuations between 116 and 122 hPa with fast changes in ambient pressure, but discounting these episodes, remains between 119.5 and 120.5 hPa.





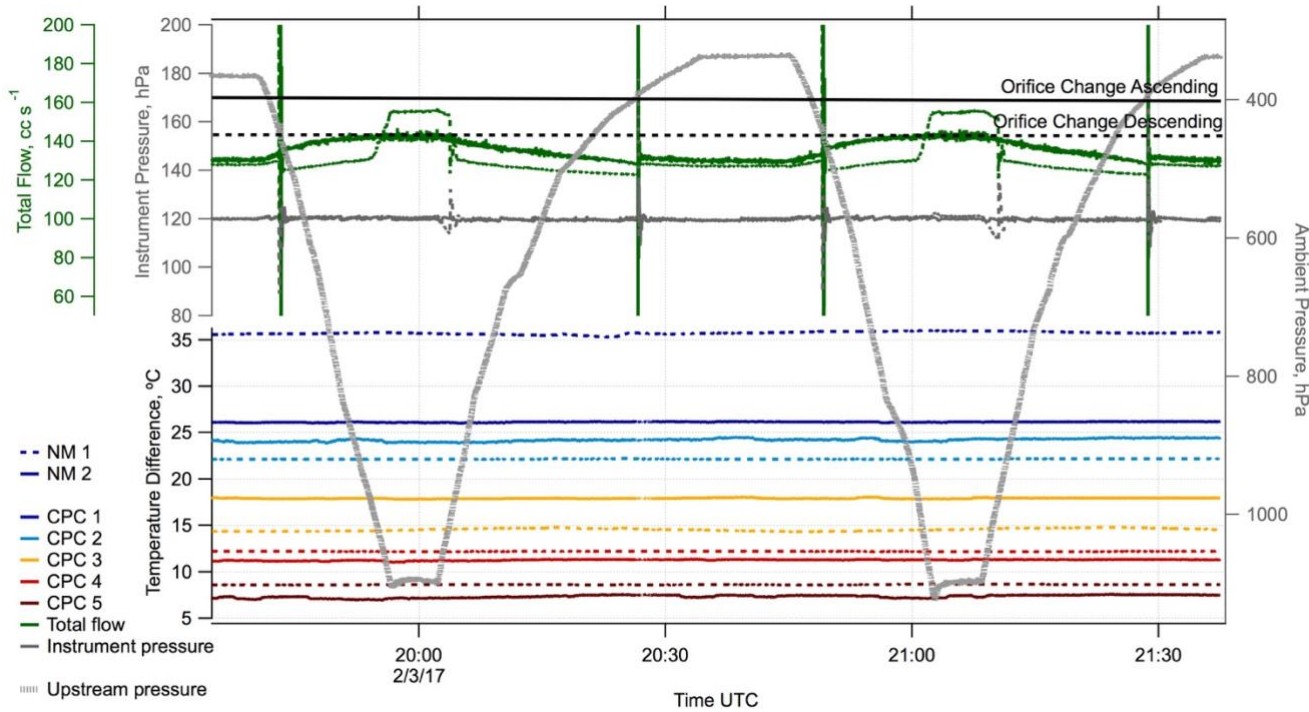

**Fig. 11 Flows, temperatures, and pressures measured by both NMASSes over the total course of an ATom flight. Ambient pressure is shown in grey, coloured lines on the bottom panel are the temperature difference between condenser and saturator in each of the CPCs. Top panel shows the total NMASS flow (green) and instrument pressure (dark grey). Solid lines are used for NMASS 2, dashed lines for NMASS 1. Ambient pressures at which the orifice is changed are shown by the black horizontal lines. This occurs at 400 hPa on ascent (solid line) and 450 hPa on descent (dashed line). The spikes in instrument pressure and total flow are a response to the orifice change. At inlet pressures >960 hPa, NMASS-1 total flows could not be maintained with acceptable limits in ATom-1 and ATom-2; this has been fixed for subsequent deployments.**

## 6.2 Example data from the ATom mission

Example data from the NMASS measuring during ATom in February 2017 are shown in Fig. 12. Concentrations of the 10 NMASS channels, along with the total concentration of particles with diameter between 63-1000 nm measured by an ultra-high sensitivity aerosol spectrometer (Kupc et al., 2017) and the inverted size distribution are shown in panel (a). From about 01:41 to 01:46 the large difference between the concentrations in channels 1 and 2 indicates recent or active new particle formation, since particles between 3 and 7 nm have a relatively short lifetime in the atmosphere and so must have been formed recently. Channels 1 and 2 vary independently of channels 3, 4 and 5 between 01:41 and 01:45, indicating 2 distinct modes of particles present. As concentrations in channels 1 and 2 become very similar after 01:48 and concentrations in channels 1-4 become very similar after 01:58, this indicates that most particles are now larger than 7 nm, and then larger than 28 nm.

Average size distributions for 1 minute of data each are shown in panels (b) through (d) of Fig 12. These illustrate recent new particle formation (b), particles growing into the Aitken mode (c) and further growth to a mode diameter around 40 nm



(d).



**Fig. 12 Example data taken on the NASA DC-8 aircraft during ATom in February 2017. The top of panel (a) shows the STP number concentrations measured by each channel of both NMASSes, as well as the total concentration of particles from 63-1000 nm measured by the UHSAS, which is used to constrain the inversion, and the ambient pressure. The bottom of panel (a) shows the inverted size distribution. Panels (b), (c) and (d) show an average of 1 minute of data (shown by the dashed vertical lines of the corresponding colors in panel (a)) at the 3 times to show different example size distributions. Recent new particle formation produced high concentrations of small particles at 01:41, as indicated by the large difference in particle number concentration between the NM1 CPC 1 and 2 in panel (a), corresponding to a mode diameter in the size distribution below 3 nm. At 01:50, most particles are in the Aitken mode, as shown by the broad spacing of the middle diameter CPC concentrations in the top panel, and**





the mode diameter around 8 nm in the lower two panels. At 01:56 (d) all of the lower diameter channels measure the same concentration, as seen in the panel (a), and the mode diameter increases to around 40 nm in the size distribution. Gaps in the data are where the aircraft flew through cloud, which can cause artefacts, and so NMASS and UHSAS data are removed here.

## 7. Conclusions

The stable, reproducible characteristic of the NMASS, demonstrated in this paper, allow measurements of fast time-response size-selected aerosol concentrations in flight over rapidly changing ambient pressure from the boundary layer to the stratosphere. For the ATom mission, two NMASS instruments were modified and extensively calibrated and tested in the laboratory with a range of particle sizes. The response function of each of ten CPC channels was determined, and the repeatability of the $d_{50}$ of each channel was determined to be better than 5% for all but one channels, which had a

repeatability of 8%. An evaluation of the propagation of all uncertainties for a range of size distributions shows that particle number, surface area, and volume concentrations within the nucleation and Aitken size range can be determined to better than 20% for typical particle size distributions. Performance may be worse for very low concentrations of particles with modes at the extreme edges of the NMASS detection range. No sensitivity in sizing performance to particle composition was found for three diverse particle compositions.

Performance in flight shows that temperatures, pressures, and flows remain within acceptable bounds except for pressures >960 hPa (an issue that has been fixed prior to the third ATom deployment). Concentrations measured by the two NMASS instruments flying in parallel are self-consistent and also consistent with measurements made with a UHSAS instrument. The two NMASS instruments flying on the ATom mission are providing a high-quality, contiguous tropospheric dataset of nucleation- and Aitken-mode size distributions with global coverage of the Pacific and Atlantic Ocean basins and seasonal

variation. These data will be used to evaluate the dominant mechanisms of atmospheric new particle formation and the contribution of nucleated particles to the global distribution of cloud-active particles and, through model sensitivity studies, their subsequent influence on radiative forcing.

### Data Availability

Calibration, laboratory testing and in-flight data are available upon request to the corresponding author. Processed and
quality-controlled data for the ATom mission are publicly available at the ATom data archive: https://dx.doi.org/10.5067/Aircraft/ATom/TraceGas_Aerosol_Global_Distribution.

### Author Contributions

All authors contributed substantially to the work presented in this paper. C. Brock, J. Wilson, D. Gesler and J. M. Reeves designed, built, programmed, and tested the NMASS instruments, which were then modified by C. Williamson and C.
Brock. C. Williamson and A. Kupc calibrated the NMASSes and collected data during ATom-1 and -2 missions. F. Erdesz and R. McLaughlin made the orifice changer system and other instrument parts. C. Williamson prepared the manuscript with contributions from all authors.



## Acknowledgements

The authors acknowledge support by NASA's Earth System Science Pathfinder Program under award NNH15AB12I and by NOAA's Health of the Atmosphere and Atmospheric Chemistry, Carbon Cycle, and Climate Programs. Agnieszka Kupc is supported by the Austrian Science Fund FWF's Erwin Schrodinger Fellowship J-3613. We would like to thank Bernadett.

Weinzierl, Maximilian Dollner, T. Paul Bui and Glenn S, Diskin for access to their preliminary data. Jose Jimenez and Pedro Campuzano-Jost kindly loaned us a TSI 3776 CPC and Paul Ziemann an electrometer. Finally, we would like to thank David Fahey, Karl Froyd, Daniel Murphy, Steven Ciciora and Daniel Law for insightful discussions.

## Disclaimer

This publication's contents do not necessarily represent the official views of the respective granting agencies. The use or

mention of commercial products or services does not represent an endorsement by the authors or by any agency.

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

**Table 1. Condenser temperatures, 50% cut-off diameters ($d_{50}$) and 50,10 and 90% counting efficiencies from the limonene ozonolysis calibration of the NMASS for a saturator temperature of 39.0 ± 0.1°C.**

| Instrument | Channel | $T_{condenser}$, °C | $d_{50}$, nm | $d_{10}$, nm | $d_{90}$, nm |
|---|---|---|---|---|---|
| 1 | 1 | 2.6 | 3.19±0.16 | 2.43±0.12 | 4.46±0.22 |
| 1 | 2 | 16.2 | 6.90±0.35 | 5.64±0.28 | 8.89±0.45 |
| 1 | 3 | 21.4 | 13.7±0.96 | 12.1±0.85 | 16.4±1.15 |
| 1 | 4 | 26.2 | 26.8±2.68 | 23.1±2.31 | 32.7±3.27 |
| 1 | 5 | 29.7 | 59.1±6.50 | 42.8±4.71 | 85.1±9.36 |
| 2 | 1 | 12.2 | 4.83±0.31 | 4.441±0.23 | 5.56±0.44 |
| 2 | 2 | 13.6 | 5.88±0.58 | 5.23±0.52 | 7.12±0.67 |
| 2 | 3 | 20.4 | 10.8±0.97 | 9.26±0.83 | 11.8±1.06 |
| 2 | 4 | 27.2 | 20.1±2.01 | 18.1±1.81 | 23.2±2.32 |
| 2 | 5 | 30.7 | 37.5±4.88 | 33.4±4.34 | 40.9±5.32 |



**Table 2. Three-mode lognormal descriptions of representative aerosol size distribution cases giving model number concentration $N$, geometric mean diameter $D_g$, and geometric standard deviation $\sigma_g$. Mean relative bias $B_Y$ (Eq. 3) and standard deviation $\sigma_Y$ (Eq. 2) from size distributions recovered by 1000 Monte Carlo simulations of 10-channel NMASS performance for parameters number ($Y=n$), surface area ($Y=s$) and volume ($Y=v$).**

| | Mode 1 | | | Mode 2 | | | Mode 3 | | | Number | | Surface area | | Volume | |
|---|---|---|---|---|---|---|---|---|---|---|---|---|---|---|---|
| # | $N$ | $D_g$ | $\sigma_g$ | $N$ | $D_g$ | $\sigma_g$ | $N$ | $D_g$ | $\sigma_g$ | $B_n$ | $\sigma_n$ | $B_s$ | $\sigma_s$ | $B_v$ | $\sigma_v$ |
| 1 | 1700 | 0.01 | 1.7 | 800 | 0.055 | 1.6 | 0 | NA[1] | NA | 0.08 | .054 | -0.002 | 0.056 | -0.010 | 0.062 |
| 2 | 4000 | 0.009 | 1.6 | 200 | 0.05 | 2.5 | 0 | NA | NA | 0.10 | .052 | 0.047 | 0.069 | 0.049 | 0.092 |
| 3 | 1800 | 0.011 | 1.5 | 2400 | 0.04 | 1.7 | 400 | 0.09 | 1.9 | 0.10 | .043 | 0.006 | 0.068 | 0.001 | 0.076 |
| 4 | 20 | 0.004 | 1.35 | 3 | 0.015 | 1.7 | 0 | NA | NA | 0.004 | 0.24 | 0.12 | 0.40 | 0.39 | 0.91 |
| 5 | 45 | 0.1 | 2 | 8 | 0.01 | 2.5 | 0 | NA | NA | 0.18 | 0.13 | 0.056 | 0.17 | 0.040 | 0.18 |
| 6 | 1600 | 0.055 | 1.8 | 0 | NA | NA | 0 | NA | NA | 0.15 | 0.045 | 0.003 | 0.059 | -0.007 | 0.061 |
| 7 | 1600 | 0.032 | 1.6 | 70 | 0.1 | 2 | 100 | 0.01 | 2.5 | 0.12 | 0.040 | 0.019 | 0.095 | 0.012 | 0.12 |
| 8 | 15000 | 0.0025 | 1.6 | 80 | 0.08 | 1.6 | 0 | NA | NA | -0.20 | 0.13 | -0.002 | 0.072 | 0.002 | 0.097 |

[1]NA: Not applicable