# Peer review of "Fast time response measurements of particle size distributions in the 3-60nm size range with the Nucleation Mode Aerosol Size Spectrometer"

_Atmospheric Measurement Techniques, 2018_

## Referee Comment (RC1) · Anonymous Referee #1 · 6 Mar 2018

This is an excellent paper. It nicely discusses the design of the NMASS in the context of prior work, as well as essential technical aspects of the NMASS design, function, calibration, and data inversion. It also compares measurements of the NMASS with those from an SMPS, which provides better resolution in size (but not time), and illustrates results that can be obtained with the NMASS when used on a research aircraft. The paper is also very well written. I feel it should be published with only minor editorial corrections and revisions.

Suggestions, to be included at the discretion of the authors (not essential):

1. p. 4, lines 3-5. The authors might consider mention of Winkler's DMA train among the fast-time response instruments for measuring nanoparticle size distributions (Pichelstorfer et al. 2018). Due to its higher sensitivity to number concentrations and lower weight and power requirements, the NMASS is more suitable for use on aircraft. However, the DMA train is a new development that has its place and might be mentioned.

2. p. 8, figure caption. The authors cite the Airmodus PSM (Vanhanen et al., 2011) as an instrument that uses a working fluid other than n-butanol (diethylene glycol) and a two-stage CPC detector. These aspects of the Airmodus instrument are based on the earlier work of Iida et al. (2009), who identified diethylene glycol as a suitable condensing fluid for sub 3 nm particles and pioneered the use of a butanol CPC "booster" as a second stage detector for the small droplets on which diethylene glycol had condensed. The authors should consider citing Iida's contributions as well. Iida et al. (2009) also experimentally studied differences in activation efficiencies for positively and negatively charged sub 3-nm particles. This may be pertinent to your discussion on p. 11.

3. p. 15: Although it may be obvious to those who have worked with CPCs, you might point out the reason that the counting efficiency for CPC5 decreases with decreasing size below 7 nm.

4. pages 19 & 21. The paper indicates that the method used in section 4.1 was used to invert both the NMASS and SMPS data. The discussion is mercifully concise, but I am still curious about several points:

-Are the data from the two NMASS instruments merged prior to inversion, or are the data merged separately and the inverted distributions merged after inversion? It is not entirely obvious to me which approach would be preferable, and the paper provides no insight. Systematic differences between the instruments that could lead to large errors in concentration differences between adjacent channels might argue in favor of separate inversions, but constraining a single inversion with more data points might

argue in favor of merging the data before inversion. A sentence or two would suffice.

-Standard SMPS inversion methods (e.g., TSI's AIM software as well as software used by most aerosol scientists) would lead to accurate results for mean size and concentration for aerosols sampled from a DMA, (Figs. 9 & 10). However, the size distributions provided by those methods would be broader than the measured size distributions. This is because the distributions delivered by the DMA are broad relative to the transfer function of the DMA in the SMPS. I believe the modified Twomey technique that was used in this analysis should not suffer from that problem, but too few details are given for me to be certain. Nevertheless, if the DMA in Fig. 9 was operating properly, the size distribution of the sampled aerosol would be exactly equal to the DMA transfer function times the size distribution of the aerosol from the atomiser. It would be interesting to see this theoretical size distribution on Fig. 10 as well. If it agreed well with the blue dashed line, it would provide further support for the validity of your inversion algorithm. (Most aerosol scientists who have worked extensively with data inversion -I am certainly among them- are skeptical about the results.)

Minor Editorial Changes:

p. 7 line 14: missing ")." following Hanson et al., 2002) p. 11, line 9: and mostly limited to the smallest... p. 13 line 14: Should this be Fig. 5, not Fig. 6? p. 16, caption to Fig 7: "smallest cut-off sizes by." ??? p. 27, line 9: "for all but one channel"

Iida, K., M. R. Stolzenburg and P. H. McMurry (2009). "Effect of Working Fluid on Sub-2 nm Particle Detection with a Laminar Flow Ultrafine Condensation Particle Counter." Aerosol Science and Technology 43(1): 81-96. Pichelstorfer, L., D. Stolzenburg, J. Ortega, T. Karl, H. Kokkola, A. Laakso, K. Lehtinen, J. N. Smith, P. H. McMurry and P. M. Winkler (2018). "Resolving nanoparticle growth mechanisms from size and time dependent growth rate analysis." Atmospheric Chemistry & Physics 18: 1307-1323. doi: 10.5194/acp-18-1307-2018.

---

## Referee Comment (RC2) · Anonymous Referee #2 · 12 Mar 2018

The present manuscript by Williamson et al. presents a very exhaustive description and characterization of two nucleation mode aerosol size spectrometer (NMASS) units, which consists of five CPCs operated in parallel and at constant low pressure. The NMASS sizing, inversion and field performance is verified experimentally and the error sources are investigated thoroughly. The manuscript is well written and the experiments and data analysis are properly conducted. I have only very minor comments on the manuscript, and suggest its publication in AMT.

P4 l3-13, an SMPS has been shown to work quite well down to 3 s scan time (Trostl et

al. 2015).

P13 l15-19, how is the theoretical Kelvin diameter estimated for Fig3? Does it take into account the flow velocity and supersaturation profiles inside the condenser? Check for example Giechaskiel et al. (2011). This should be discussed a little bit more in the main text, since the disagreement in fig3 is quite large

Giechaskiel, B., Wang, X., Gilliland, D., Drossinos, Y. (2011). The effect of particle chemical composition on the activation probability in n-butanol condensation particle counters. J Aerosol Sci 42:20-37.

Trostl, J., Tritscher, T., Bischof, O. F., Horn, H. G., Krinke, T., Baltensperger, U., Gysel, M. (2015). Fast and precise measurement in the sub-20 nm size range using a Scanning Mobility Particle Sizer. J Aerosol Sci 87:75-87.

---

## Referee Comment (RC3) · Anonymous Referee #3 · 26 Mar 2018

Overall comments: This is a very well-written paper that provides a detailed description of the Nucleation-Mode Aerosol Size Spectrometer (NMASS) instruments that were developed in the late 1990s and have subsequently been used successfully to measure high-temporal-resolution aerosol size distributions aboard a variety of research aircraft including the NASA ER-2 and DC-8. The authors do a very good and thorough job of describing the NMASS design, performance characteristics and calibration procedures and of carefully determining the precision and possible errors in their derived size distributions. I think the study and paper are excellent and should be published after a few

minor revisions.

Specific Comments: Figures 1 and 2: I don't understand how flow velocity is maintained constant in each of the CPCs. I had assumed that critical flow orifices (CFOs) were installed on the exhaust of each unit (as is done in standard TSI3010 or TSI3772 counters) and that system pressure was controlled by varying flow through a bypass line. However, although it shows a solenoid control valve dangling in space above the column of counters, Figure 1 seems to indicate that system pressure is controlled using the valve located just upstream of pump; varying flow through this valve would result in varying the flow velocity through the entire bank of counters. Can you please clarify? If the flow through each CPC isn't controlled by a CFO or pump, how do you know that sample flow is split equally between the five CPCs? Do you measure total flow through each counter?

The NMASS CPC design is similar to that of a TSI3025, which includes a valve to regulate the ratio of aerosol sample to sheath flow; how is the flow split maintained in the NMASS CPCs? If the ratio is dictated by flow resistances instead of valves, how does it change as the filters get dirty? How sensitive is the CPC detection efficiency to variations in flow velocity through the condenser region?

Is Fluorinert hygroscopic? If so, how does the CPC performance change over time as $H_2O$ is absorbed into the working fluid? During ATOM, the aircraft often flies through very humid regions—do you dry the sample flow and if not, how often do you change the fluid in the counters?

Page 9, orifice discussion: Sample temperature will drop as flow is expanded across the orifice, which in very humid cases, may lead to vapor condensation and associated particle losses. Was this effect considered in your experiments and analyses?

Page 9, line 30: the by pass line is not clearly indicated in Figure 1.

Figure 7: the caption is cut off and thus doesn't make complete sense.

Page 20-21 discussion on NPF formation and growth: you might mention Rodney Weber's technique of using an OPC to measure the size of droplets coming out of an ultrafine CPC to infer new particle formation and growth: Weber et al., Measurements of enhanced H2SO4 and 3-4 nm particles near a frontal cloud during the First Aerosol Characterization Experiment (ACE 1), JOURNAL OF GEOPHYSICAL RESEARCH-ATMOSPHERES , 106 (D20), 24107-24117, 2001.

Regarding use of SMPS instruments to study NPF from aircraft: even if they could be operated at fast scanning rates, standard SMPS systems lack sensitivity to nucleation mode particles at low pressure because of the particles's greatly increased electrical mobility. For example, at 100 hPa, a 10 nm particle has about the same electrical mobility as a 3 nm particle at sea level pressure.

Page 22, line 14: should be "than" instead of "then"

Page 22, lines 17-22: repeated text

---

## Author Comment (AC1) · 26 Apr 2018

**Authors' response to reviewer 1 comments on manuscript titled "Fast time response measurements of particle size distributions in the 3-60nm size range with the Nucleation Mode Aerosols Size Spectrometer", submitted to AMT 24th January 2018**

The authors would like to thank the reviewer for their considered and positive evaluations of the manuscript. Our responses are detailed below, with the reviewer comments in normal text and our response in italics.

1.  p. 4, lines 3-5. The authors might consider mention of Winkler's DMA train among the fast-time response instruments for measuring nanoparticle size distributions (Pichelstorfer et al. 2018). Due to its higher sensitivity to number concentrations and lower weight and power requirements, the NMASS is more suitable for use on aircraft. However, the DMA train is a new development that has its place and might be mentioned.

    *This is an excellent point, and the DMA train has now been considered on P4 lines 6-8. Text now reads: "A "DMA-train" composed of 6 differential mobility analyzers measures only the aerosol fraction that is charged in an ionizer (Stolzenburg et al., 2017), and is neither compact nor optimized for operation at reduced pressures."*
    *We chose to reference Stozenburg 2017, instead of the recommended Pichelstorfer 2018, as Stolzenberg 2017 provides a more in-depth description of the instrument and its capabilities.*

2.  p. 8, figure caption. The authors cite the Airmodus PSM (Vanhanen et al., 2011) as an instrument that uses a working fluid other than n-butanol (diethylene glycol) and a two-stage CPC detector. These aspects of the Airmodus instrument are based on the earlier work of Iida et al. (2009), who identified diethylene glycol as a suitable condensing fluid for sub 3 nm particles and pioneered the use of a butanol CPC "booster" as a second stage detector for the small droplets on which diethylene glycol had condensed. The authors should consider citing Iida's contributions as well. Iida et al. (2009) also experimentally studied differences in activation efficiencies for positively and negatively charged sub 3-nm particles. This may be pertinent to your discussion on p. 11.

    *Iida et al reference added to Fig 3. Caption. Caption now reads:*
    ***Fig.1 Kelvin diameter, or critical diameter, $D^*$ as a function of difference in temperature between saturator and condenser, for n-butanol (used in many commercial CPCs such as the TSI-3776 (Hermann et al., 2007)), diethylene glycol (used in commercial and custom-built two-stage CPCs such as the Airmodus Particle Size Magnifier (Vanhanen et al., 2011;Iida et al., 2009)) and Fluorinert FC-43 (used in the NMASS CPCs). The saturator temperature is 34.8 °C. For a given $D^*$ the slope of the curve for FC-43 is less than for n-butanol or diethylene glycol. The measured diameter of 50% detection efficiency, $d_{50}$, for an NMASS CPC is also shown as a function of temperature, as discussed in section 3.2. The NMASS CPC $d_{50}$s are larger than the theoretical Kelvin diameter for Fluorinert because heat and mass transfer within the condenser limits the supersaturation achieved to values less than the theoretical maximum. The range of $d_{50}$s shown here, around 40-60nm, are on the steep part of the diameter curve. This limits the largest $d_{50}$ that can be achieved with the NMASS because, in this region, a small variation in temperature difference causes a large variation in $d_{50}$, making the detection efficiency unstable.***

3.  p. 15: Although it may be obvious to those who have worked with CPCs, you might point out the reason that the counting efficiency for CPC5 decreases with decreasing size below 7 nm.

    *We assume that the referee meant 70 nm, and have added clarification to the caption of Figure 6. Text now includes "At diameters <70 nm, the roll-off in detection efficiency of CPC5 ($dp_{50}$=59.1±6.5 nm) is already evident."*

4.  pages 19 & 21. The paper indicates that the method used in section 4.1 was used to invert both the NMASS and SMPS data. The discussion is mercifully concise, but I

am still curious about several points:

-Are the data from the two NMASS instruments merged prior to inversion, or are the data merged separately and the inverted distributions merged after inversion? It is not entirely obvious to me which approach would be preferable, and the paper provides no insight. Systematic differences between the instruments that could lead to large errors in concentration differences between adjacent channels might argue in favor of separate inversions, but constraining a single inversion with more data points might argue in favor of merging the data before inversion. A sentence or two would suffice.

*This was indeed not explicitly addressed, so a sentence explaining the choice of a single inversion over 10 channels and why there are no large systematic differences between the instruments that might make this problematic has been added p20 lines 5-6. Text now reads "We use channels from both NMASSes in a single inversion and the calibrations ensure no large systematic differences between the two instruments."*

-Standard SMPS inversion methods (e.g., TSI's AIM software as well as software used by most aerosol scientists) would lead to accurate results for mean size and concentration for aerosols sampled from a DMA, (Figs. 9 & 10). However, the size distributions provided by those methods would be broader than the measured size distributions. This is because the distributions delivered by the DMA are broad relative to the transfer function of the DMA in the SMPS. I believe the modified Twomey technique that was used in this analysis should not suffer from that problem, but too few details are given for me to be certain. Nevertheless, if the DMA in Fig. 9 was operating properly, the size distribution of the sampled aerosol would be exactly equal to the DMA transfer function times the size distribution of the aerosol from the atomiser. It would be interesting to see this theoretical size distribution on Fig. 10 as well. If it agreed well with the blue dashed line, it would provide further support for the validity of your inversion algorithm. (Most aerosol scientists who have worked extensively with data inversion -I am certainly among them- are skeptical about the results.)

*We have addressed these concerns in an extra section in the supplementary material (section C) including figure S3, referenced on page 22 lines 22-23. New section with extra figure is as follows:*

**C. Verification of custom-built DMA and nano-SMPS performance**

*A custom-built DMA was used to generate the calibration aerosol used for comparison between a nano-SMPS and the NMASS in Fig. 10. The performance of this DMA was verified by atomizing a nearly monodisperse polystyrene latex (PSL) sphere aerosol with a peak diameter of 152 ± 5 nm and a polydispersity of 2.1% (ThermoFisher Scientific Series 3000 nanospheres). A CPC measured the concentration of particles exiting the DMA as the voltage was manually stepped across the peak in the transmission function (Fig. S3). A fitted Gaussian curve gives a peak diameter of 151.3 nm and a full-width at half-max (FWHM) of 17.4 nm (11.4%). The fitted peak diameter agree with the PSL size standards within uncertainties. The FWHM of the distribution is very close to the expected FWHM of 16.9 nm (11.1%) calculated from DMA theory (Knutson and Whitby, 1975) with the 10:1 sheath/aerosol flow ratio used and accounting for the polydispersity of the PSL. Thus the custom-built DMA is working close to theoretically optimal performance.*

*The custom-built DMA was used to produce a size-classified, atomized ammonium sulfate aerosol that was tested by the nano-SMPS and the NMASS (Fig. 10). The SMPS size distribution, inverted using the same Markowsky-*

Twomey algorithm that is also applied to the NMASS data, displays a FWHM of 13.6% and 12.6% for the 20 and 30 nm sizes selected, respectively. Thus the Markowsky-Twomey inversion applied to the nano-SMPS slightly broadens the aerosol generated by the custom-built DMA. This is not unexpected because the inversion applies a smoothing step, as discussed in Section 4.1.

[Figure]

*Figure S3. Concentration of particles produced by atomizing nearly monodisperse particles, classifying them in a custom-built DMA, and counting them with a CPC.*

Minor Editorial Changes:
p. 7 line 14: missing ").” following Hanson et al., 2002) - *addressed*
p. 11, line 9: and mostly limited to the smallest... - *addressed*
p. 13 line 14: Should this be Fig. 5, not Fig. 6? - *Yes, addressed*

p. 16, caption to Fig7: "smallest cut-off sizes by." ??? - *should have read "atomizing ammonium sulphate or dioctyl sebacate' – this has been added*
p. 27, line 9: "for all but one channel" - *addressed*

**References:**

Hermann, M., Wehner, B., Bischof, O., Han, H. S., Krinke, T., Liu, W., Zerrath, A., and Wiedensohler, A.: Particle counting efficiencies of new TSI condensation particle counters, J. Aerosol Sci, 38, 674-682, 2007.

Iida, K., Stolzenburg, M. R., and McMurry, P. H.: Effect of Working Fluid on Sub-2 nm Particle Detection with a Laminar Flow Ultrafine Condensation Particle Counter, Aerosol Sci. Technol., 43, 81-96, 10.1080/02786820802488194, 2009.

E.O. Knutson, E. O., and Whitby, K. T.: Aerosol classification by electric mobility: apparatus, theory, and applications, J. Aerosol Sci., 6, 443-451, doi:10.1016/0021-8502(75)90060-9, 1975.

Stolzenburg, M. R., and Mcmurry, P. H.: An Ultrafine Aerosol Condensation Nucleus Counter, Aerosol Sci. Technol., 14, 48-65, Doi 10.1080/02786829108959470, 1991.

Vanhanen, J., Mikkila, J., Lehtipalo, K., Sipila, M., Manninen, H. E., Siivola, E., Petaja, T., and Kulmala, M.: Particle Size Magnifier for Nano-CN Detection, Aerosol Sci. Technol., 45, 533-542, 10.1080/02786826.2010.547889, 2011.

---

## Author Comment (AC2) · 26 Apr 2018

**Authors' response to reviewer 2 comments on manuscript titled "Fast time response measurements of particle size distributions in the 3-60nm size range with the Nucleation Mode Aerosols Size Spectrometer", submitted to AMT 24th January 2018**

The authors would like to thank the reviewer for their considered and positive evaluations of the manuscript. Our responses are detailed below, with the reviewer comments in normal text and our response in italics.

1. P4 l3-13, an SMPS has been shown to work quite well down to 3 s scan time (Trostl et al. 2015).

   *A good point, this has now been discussed p4 lines 15-18. New text reads,*
   *"It has been shown that an SMPS performs well with scan times as low as 3s (Trostl et al., 2015), however, operation with these fast scans is challenging and uncommon, and the low charging efficiencies for nucleation and Aitken mode particles limits the sensitivity. Further, at reduced pressure, the sizing range of an SMPS may be limited because particles have higher electrical mobility at a given voltage setting."*

2. P13 lines 15-19, how is the theoretical Kelvin diameter estimated for Fig3? Does it take into account the flow velocity and supersaturation profiles inside the condenser? Check for example Giechaskiel et al. (2011). This should be discussed a little bit more in the main text, since the disagreement in fig3 is quite large.

   *The theoretical Kelvin diameter is estimated following the method in Baron and Willeke (2001) without taking into account flow velocity and supersaturation profiles. We have now included this information in the main text (p14 lines 5-11) and discussed the implications there. New text reads,*
   *"For a given temperature difference between saturator and condenser, the measured $d_{50}$ in Fig.3 is larger than the theoretical Kelvin diameter (Baron and Willeke, 2001). The Kelvin diameter is the minimum diameter at which it is possible for particles to nucleate, while $d_{50}$ is the diameter at which 50% of particles are actually detected in the instrument. The discrepancy between the theoretical Kelvin diameter and $d_{50}$ in Fig.3 is likely because the NMASS saturator does not reach the maximum theoretical supersaturation. Because we lack information on the mass and thermal diffusivities of FC-43, we cannot simulate the coupled heat and mass transfer within the condenser to explore this difference. However, as long as the degree of saturation is constant (which it is expected to be since pressure, flow and temperature are constant), the $d_{50}$ of each NMASS channel should also be constant."*

   *References:*

   Baron, P. A., and Willeke, K.: Aerosol measurement : principles, techniques, and applications, 2nd ed., Wiley, New York, xxiii, 1131 p. pp., 2001.

   Trostl, J., Tritscher, T., Bischof, O. F., Horn, H. G., Krinke, T., Baltensperger, U., and Gysel, M.: Fast and precise measurement in the sub-20 nm size range using a Scanning Mobility Particle Sizer, J. Aerosol Sci, 87, 75-87, 10.1016/j.jaerosci.2015.04.001, 2015.

---

## Author Comment (AC3) · 26 Apr 2018

**Authors' response to reviewer 2 comments on manuscript titled "Fast time response measurements of particle size distributions in the 3-60nm size range with the Nucleation Mode Aerosols Size Spectrometer", submitted to AMT 24th January 2018**

The authors would like to thank the reviewer for their considered and positive evaluations of the manuscript. Our responses are detailed below, with the reviewer comments in normal text and our response in italics.

1. Figures 1 and 2: I don't understand how flow velocity is maintained constant in each of the CPCs. I had assumed that critical flow orifices (CFOs) were installed on the exhaust of each unit (as is done in standard TSI3010 or TSI3772 counters) and that system pressure was controlled by varying flow through a bypass line. However, although it shows a solenoid control valve dangling in space above the column of counters, Figure 1 seems to indicate that system pressure is controlled using the valve located just upstream of pump; varying flow through this valve would result in varying the flow velocity through the entire bank of counters. Can you please clarify? If the flow through each CPC isn't controlled by a CFO or pump, how do you know that sample flow is split equally between the five CPCs? Do you measure total flow through each counter?
The NMASS CPC design is similar to that of a TSI3025, which includes a valve to regulate the ratio of aerosol sample to sheath flow; how is the flow split maintained in the NMASS CPCs? If the ratio is dictated by flow resistances instead of valves, how does it change as the filters get dirty? How sensitive is the CPC detection efficiency to variations in flow velocity through the condenser region?

   *The flow through the CPCs is determined by the pressure drop across the CPC saturator filter (see fig 2) and the solenoid valve on the bypass flow. The pressure drop across the capillary of each CPC is measured continuously. Calibrations were done in the lab to relate the pressure drop to the flow through the capillary and this used to calculate the flow for each CPC continuously during operation. This was indeed unclear in the original manuscript and a full explanation has now been added (p9 line 32 to p10 line 2). The new text reads:*
   *"The flow through each CPC is determined by the pressure drop across the filter in the saturator (see Fig. 1)*

   *and the proportional control valve. The pressure drop across each capillary is continuously measured*

   *during operation, as shown in Fig. 1. Calibrations were done to relate these pressure drops to a volumetric*

   *flow, and it is these flows that are then used to determine the concentration in each channel from the*

   *number of particles counted."*

   *The flow across each CPC capillary is constantly monitored. If the flow at a given upstream pressure drops by above 10%, the filter is changed to avoid possible effects on the CPC detection efficiency.*

   *The line connecting the solenoid control valve to the pump was indeed missing in this figure, as the review later points out. We have replaced this in a new version of Fig 1 (see below).*

[Figure]

2. Is Fluorinert hygroscopic? If so, how does the CPC performance change over time as H2O is absorbed into the working fluid? During ATOM, the aircraft often flies through very humid regions do you dry the sample flow and if not, how often do you change the fluid in the counters?

*Flow into the NMASS is always dried to below 20% RH, so we do not expect problems from $H_2O$ being absorbed in the working fluid. This explanation was missing in the original manuscript, and so has been added to p25 lines 22-25, new text reads:*
*"For operation on ATom the sample flow is passed through a large diameter Nafion$^{TM}$ dryer before entering the NMASSes. This reduced the relative humidity to below 20%. This ensures that particles measured in the NMASS are classified consistently by dry diameter, and avoids potential problems of particle losses associated with water vapour condensation during flow expansion in the orifice or effects of water vapour on the performance of the CPC working fluid."*

*.*

3. Page 9, orifice discussion: Sample temperature will drop as flow is expanded across the orifice, which in very humid cases, may lead to vapor condensation and associated particle losses. Was this effect considered in your experiments and analyses?
*The sample flow is dried before entering the instrument (this explanation was missing from the original manuscript and has now been added to p25 lines 22-25, and page 9 lines 30-31), so the problem of vapor condensation and associated particle losses as the flow expands across the orifice can be discounted as we*

*do not experience high humidities in the instrument. For operation without a drier in humid areas this would need to be considered. New text reads:*
*Page 9: "A Nafion drier upstream of the NMASS instrument maintains RH to <20%, elminating the possibility of condensation in the pressure reducer."*

*Page 25: "For operation on ATom the sample flow is passed through a large diameter nafion$^{TM}$ dryer before entering the NMASSes. This reduced the relative humidity to below 20%. This ensures that particles measured in the NMASS are classified consistently by dry diameter,and avoids potential problems of particle losses associated with water vapour condensation during flow expansion in the orifice or effects of water vapour on the performance of the CPC working fluid."*

4.  Page 9, line 30: the by pass line is not clearly indicated in Figure 1.
    *This bypass line was missing in figure 1 and has now been corrected (see above)*

5.  Figure 7: the caption is cut off and thus doesn't make complete sense.
    *This has now been corrected – figure caption now reads:*
    **Fig. 7 Counting efficiency of NMASS 1 as a function of particle diameter for particles of different chemical composition: limonene ozonolysis products (diamonds), atomized ammonium sulphate (stars) and dioctyl sebacate (circles). Only three channels are shown here because it was not possible to produce atomized particles small enough for the two channels with the smallest cut-off sizes by atomizing ammonium sulphate or dioctyl sebacate. Counting efficiencies fall with decreasing particle diameter as particles become smaller than the Kelvin activation diameter of each channel. There is no statistically significant sensitivity of counting efficiency to particle composition.**

6.  Page 20-21 discussion on NPF formation and growth: you might mention Rodney Weber's technique of using an OPC to measure the size of droplets coming out of an ultrafine CPC to infer new particle formation and growth: Weber et al., Measurements of enhanced H2SO4 and 3-4 nm particles near a frontal cloud during the First Aerosol Characterization Experiment (ACE 1), JOURNAL OF GEOPHYSICAL RESEARCHATMOSPHERES, 106 (D20), 24107-24117, 2001.
    *We have now included this in our summary of airborne aerosol size distributions measurements on page 4 lines 9-10. New text reads*
    *"Weber et al. (2001) made airborne measurements using an optical particle counter as the sensor for an ultra-fine CPC to determine the size of the grown droplets and infer the size distribution of 3-10 nm particles that nucleated them."*

7.  Regarding use of SMPS instruments to study NPF from aircraft: even if they could be operated at fast scanning rates, standard SMPS systems lack sensitivity to nucleation mode particles at low pressure because of the particles's greatly increased electrical mobility. For example, at 100 hPa, a 10 nm particle has about the same electrical mobility as a 3 nm particle at sea level pressure.

    *This is an excellent point, which we had neglected in the original manuscript. We have now included it in the discussion of SMPS performance for airborne measurements on page 4 lines 13-14. Text reads:*
    *"Weber et al. (2001) made airborne measurements using an optical particle counter as the sensor for an ultra-fine CPC to determine the size of the grown droplets and infer the size distribution of 3-10 nm particles that nucleated them. "*

8.  Page 22, line 14: should be "than" instead of "then"
    *addressed*

9.  Page 22, lines 17-22: repeated text
    *addressed*